# On the Faithfulness of Visual Thinking: Measurement and Enhancement

## Abstract

Recent large vision–language models (LVLMs) can generate vision–text multi-modal chain-of-thought (MCoT) traces after reinforcement fine-tuning (RFT). However, we observe that the visual information incorporated in MCoT is often inaccurate (though it may potentially yield correct answers). This phenomenon indicates a lack of faithfulness in the vision part of the MCoT reasoning. We attribute this unfaithfulness to *the RL reward design in RFT, which solely incentivizes the format of interleaved vision-text cues*. That is, it encourages the model to incorporate visual information into its text reasoning steps without considering the correctness of the visual information. In this paper, we first probe the faithfulness of MCoT by measuring how much the prediction changes when its visual and textual thoughts are intervened. Surprisingly, the model's predictions remain nearly unchanged under visual intervention but change significantly under textual intervention, indicating that **the visual evidence is largely ignored**. To further analyze the visual information, we introduce a novel and automated LVLM-based evaluation metric that quantifies the faithfulness of visual cues from two perspectives: reliability and sufficiency. Our evaluation reveals that the visual information in current MCoT traces is simultaneously unreliable and insufficient. To address this issue, we propose a novel MCoT learning strategy termed Sufficient-Component Cause Model (SCCM) learning. This approach encourages the MCoT to generate sufficient yet minimal visual components that are **independently capable of leading to the correct answer**. We note that the proposed SCCM is annotation-free and compatible with various RFT for MCoT in a plug-and-play manner. Empirical results demonstrate that SCCM consistently improves the visual faithfulness across a suite of fine-grained perception and reasoning benchmarks.

## 1 Introduction

Multimodal Chain-of-Thought (MCoT) reasoning marks a pivotal advancement in the capabilities of Large Vision-Language Models (LVLMs), specifically enhancing the interpretability and intuitiveness of their reasoning processes for human users (Wang et al., 2025c). Unlike conventional text-only Chain-of-Thought (CoT) approaches (Wei et al., 2022; Team et al., 2025; Guo et al., 2025b), vision–text MCoT fundamentally integrates the visual modality into the reasoning pathway. This paradigm closely mirrors human cognition, which inherently fuses visual and linguistic information (Baddeley, 2012; Paivio, 2013). By grounding reasoning in both visual and textual contexts, MCoT provides LVLMs with a more transparent and relatable cognitive process, making complex model outputs significantly more accessible and understandable.

Recent breakthroughs have further demonstrated the potential of MCoT following the "thinking with images" paradigm (Su et al., 2025c; Hu et al., 2024; Su et al., 2025b; OpenAI, 2025). A promising direction of involving "image thoughts" in reasoning is to utilize the profound visual grounding ability of the pretrained model, by encapsulating it in an image zoom-in tool, enabling it to actively gather information from the original images by calling tool functions in an *agentic paradigm* (Plaat et al., 2025; Qian et al., 2025). This design facilitates reinforcement fine-tuning (RFT) (Schulman et al., 2017; Shao et al., 2024) in an agentic manner (Hu et al., 2024; Li et al., 2025), which has been widely verified to significantly improve the tool calling ability in LLM, *e.g.*, RAG (Jin et al., 2025) and AI Agent (Luo et al., 2025), *et al*. Based on this, such work represented by DeepEyes (Zheng

**1. Irrelevant Visual Information**

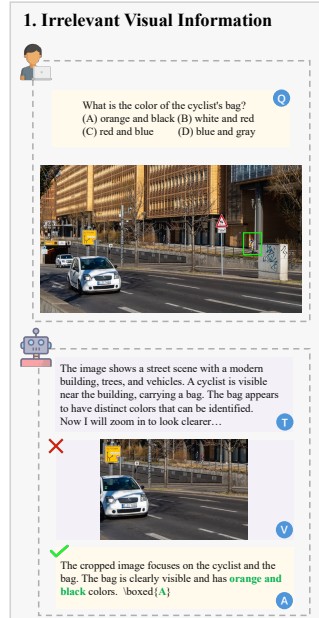

**2. Insufficient Visual Information**

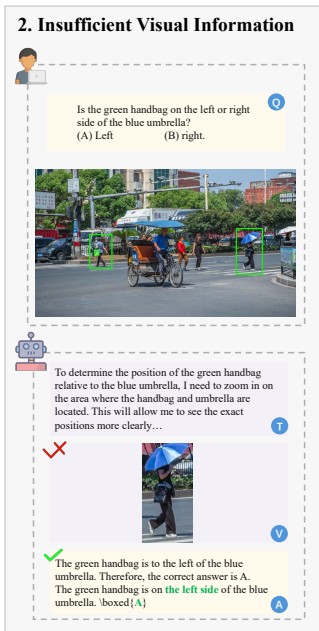

**3. Incorrect Perception**

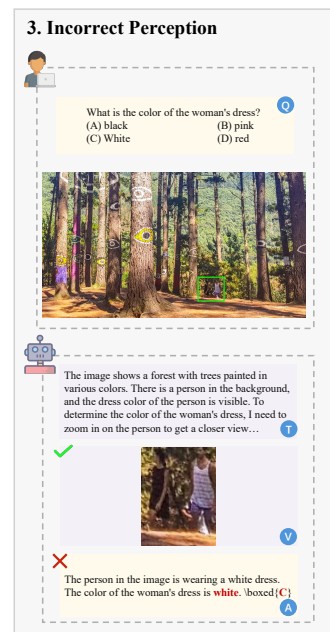

Figure 1: The mistakes present in the MCoT generated by current works (Zheng et al., 2025; Su et al., 2025a) on *V\** Bench dataset, can be divided into three categories: 1) irrelevant visual information; 2) incomplete and insufficient visual information; 3) incorrect perception.

et al., 2025) and Pixel-Reasoner (Su et al., 2025a) has achieved promising performance on various fine-grained perception and reasoning benchmarks like *V\** Bench (Wu & Xie, 2024).

However, we found that there are obvious mistakes in the generated MCoT by case study on *V\** Bench dataset, primarily divided into three categories: 1) irrelevant visual information; 2) incomplete and insufficient visual information; 3) incorrect perception. Cases are illustrated in Figure 1, which shows that inaccurate and insufficient visual information in MCoT may still yield definite, even accurate, answers, suggesting that **the MCoT can be unfaithful**. We attribute this unfaithfulness to the most widely used RL reward design (Zheng et al., 2025; Su et al., 2025a) *which only encourages the **presence** rather than the **correctness and sufficiency** of interleaved visual cues*. As a result, it can be easily hacked by introducing arbitrarily ineffective visual cues without query-related content, and deriving the final answer based solely on the textual reasoning. Such issues are more pronounced when given easy queries, where additional visual cues often offer limited benefit.

Above analysis motivates us to make an in-depth evaluation of the faithfulness of MCoT. Specifically, we first probe the faithfulness of MCoT through intervention (Hagmayer et al., 2007) on its visual and textual parts, respectively, measuring how much the prediction changes when its visual and textual thoughts are corrupted. Notably, the model's predictions remain nearly unchanged under visual intervention but change severely with textual intervention, indicating that *visual evidence can be largely ignored and thus contributes less to the model's predictions than textual evidence*. To further diagnose the visual information in MCoT, we introduce an automated LVLM-based evaluation pipeline that quantifies faithfulness from two perspectives: reliability and sufficiency. Specifically, with an external LVLM as a judger, 1) it determines whether the input visual components are *reliable* for the model's prediction; and 2) for *sufficiency*, it judges whether the input visual components can correctly answer the user's query. We conducted extensive evaluations to assess the visual faithfulness of MCoT generated by representative multimodal reasoning models (Zheng et al., 2025; Su et al., 2025a), which reveals that the visual components in MCoT are oftentimes less reliable and insufficient for correct answers, which might be even unrelated to the model's final predictions.

To address this issue, we propose Sufficient-Component Cause Model (SCCM) learning (Rothman, 1976; Flanders, 2006), in which we force the visual components to be *sufficient-and-minimal* for correct answers, *i.e.*, 1) the correct answer can be derived *solely* from the visual components of MCoT, and 2) the visual components contain no extra information that is unrelated to correct answers. This design further offers key advantages: 1) it encourages robust visual reasoning by requiring visual evidence to independently yield correct answers, thereby avoiding excessive reliance on textual rea-

soning that bypasses visual reasoning; 2) it enhances MCoT faithfulness by ensuring the correctness of visual cues, leading to rigorous visual reasoning; and 3) it facilitates a more traceable reasoning process and provides a more intuitive understanding of predictions.

The proposed SCCM is annotation-free and compatible with various RFT training for MCoT, which consistently improves faithfulness metrics across a range of fine-grained perception and reasoning benchmarks. Our main contributions include:

- We reveal the problem of unfaithfulness of visual-text MCoT where *visual evidence is largely ignored*, and introduce an evaluation pipeline to quantify the faithfulness of MCoT.
- We propose Sufficient-Component Cause Model (SCCM) learning, a simple and effective reward modeling mechanism that enhances the multi-modal reasoning ability by improving the faithfulness of the MCoT.

## 2 RELATED WORK

**Vision-language Models Reasoning.** Chain-of-Thought (CoT) (Wei et al., 2022) has been widely recognized as a key technology for enhancing the reasoning capabilities of large language models (LLMs). Inspired by the success of Guo et al. (2025a), researchers are actively exploring the application of similar reinforcement learning approaches to large vision–language models (LVLMs) (Peng et al., 2025; Zhang et al., 2025; Liu et al., 2025). Wang et al. (2025a) propose textual bounding boxes as traceable evidence to enhance visual grounding reasoning, and leverages high-quality bounding box annotations for IoU reward in reinforcement learning. However, existing approaches primarily focus on text-only reasoning and have not yet fully explored the distinctive reasoning paradigms that LVLMs may support, *e.g.*, incorporating visual evidence explicitly into the reasoning process.

**Thinking with Image.** Unlike text-only reasoning that treats vision as a static, initial context (Su et al., 2025c), the "thinking with images" paradigm actively leverage visual information as intermediate steps in the reasoning process, through extrinsic operation, *e.g.*, toolkit and code executor (Shen et al., 2024; Su et al., 2025b; Zheng et al., 2025; OpenAI, 2025) and instrinsic generation or imagination (Chern et al., 2025; Xu et al., 2025). A promising paradigm is involving visual information in an agentic manner (Zheng et al., 2025; Su et al., 2025a), which gathers information from the original images by tool calling, such as zoom-in tool. Despite these initial advances, the validity and reliability of such visual reasoning paradigms remain underexplored.

**Reasoning Faithfulness.** Faithfulness is formally defined as how well the stated explanation accurately reflects the actual reasoning process of the model. It has received sustained research attention in LLMs (Bao et al., 2024; Tanneru et al., 2024; Paul et al., 2024), and its evaluation is non-trivial, due to the large parameter scale and the black-box nature of LLMs (Agarwal et al., 2024). Lanham et al. (2023) apply different interventions to the CoT and observe the resulting changes in its final answers to evaluation CoT faithfulness. Xiong et al. (2025) utilize counterfactual intervention to investigate the faithfulness of the reasoning process. However, these methods primarily focus on textual CoT in LLMs, leaving the faithfulness of reasoning in LVLMs, particularly in their distinctive paradigms such as "thinking with images", largely unexplored. It is further complicated by the need of LVLMs for visual perception beyond text (Yu et al., 2025).

## 3 PRELIMINARY

**Definition of Multimodal Chain-of-Thought (MCoT).** We note that in the agentic "thinking with images" paradigm with visual grounding and tool calling (Zheng et al., 2025; Su et al., 2025a), the visual information is incorporated via image zoom-in tool calls rather than being generated by the model. Thus, the visual information can be regarded as observation tokens, which are appended to the ongoing reasoning process to guide subsequent MCoT. For Visual Question-Answering tasks, given an input image $I$ and a user query $Q$, the agentic multimodal reasoning process can be formulated as:

$$\mathbf{y} = \{(T_0, V_0), (T_1, V_1), ..., (T_t, V_t), A \mid I, Q\} \tag{1}$$

where $\mathbf{T} = \{T_1, T_2, ..., T_t\}$ and $\mathbf{V} = \{V_1, V_2, ..., V_t\}$ represent the textual and visual reasoning steps respectively, and $A$ is the final answer in the model's response $\mathbf{y}$. Therefore, the MCoT is defined as $\mathbf{MCoT} = \{(T_0, V_0), (T_1, V_1), ..., (T_t, V_t)\}$. Figure 2a shows the "thinking with images" paradigm.

**MCoT Faithfulness.** Faithfulness demands that the stated reasoning accurately, completely, and faithfully reflect the model's actual reasoning process, *i.e.*, it accurately represents the reasoning

process behind the model's prediction (Jacovi & Goldberg, 2020). Specifically for "thinking with images" MCoT, the faithfulness is manifested in 1) *Casual Consistency*. The textual $\mathbf{T}$ and visual $\mathbf{V}$ parts of MCoT shall have a causal relationship with the final answer $A$, rather than a fictitious association; 2) *Information Sufficiency*. Both the textual $\mathbf{T}$ and visual $\mathbf{V}$ parts of MCoT independently retain sufficient information from the input $I$ and $Q$ to derive the correct answer. Otherwise, it indicates that MCoT has fabricated or omitted information.

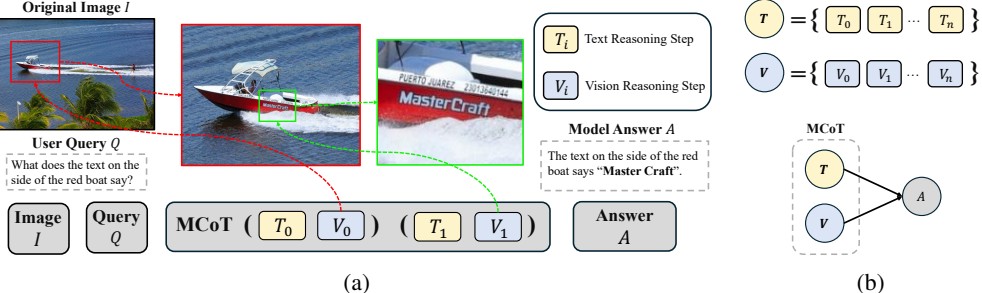

(a)                                                        (b)

Figure 2: (a) The illustration of "thinking with image" paradigm. The visual information in MCoT is introduced by zoom-in tool call as observation tokens, and appended to the ongoing reasoning process to guide subsequent steps; (b) The Structural Causal Model (SCM) of MCoT and the answer. The directed arrow between two nodes indicates a causal relationship between them, which can be verified via intervention experiments.

## 4 MCoT Faithfulness Evaluation

We introduce our MCoT faithfulness evaluation pipeline. We first probe the faithfulness of MCoT by intervention experiments for causal analysis against the textual and visual parts of MCoT and predicted answers in Sect. 4.1, revealing that the visual evidence is largely ignored. We then detail our LVLM-based evaluation pipeline that quantifies faithfulness on reliability and sufficiency of the visual components of MCoT in Sect. 4.2.

### 4.1 Causal Anaylysis

We conduct causal analysis via intervention experiments to assess the causal relationships between textual/visual components and predicted answers, by measuring how much the final predicted answer changes after intervening on textual/visual components in the generated MCoT. Figure A1 in Appendix A.1.4 shows a case of interventions on textual/visual components. This analysis adheres to the Structural Causal Model (SCM) framework (Pearl, 2009), illustrated in Figure 2b.

**Definition 1** *Average Treatment Effect (ATE). The ATE (Rubin, 1974) measures the effect of an intervention (treatment) applied to variable $X$ on an outcome variable $Y$, by comparing the expectation of $Y$ under the intervention $do(X)$ to its expectation under no intervention $X$.*

$$ATE = E(Y|do(X)) - E(Y|X) \qquad (2)$$

If the ATE resulting from an intervention on variable $X$ is significantly non-zero, it suggests that the intervention exerts an average influence on the outcome variable $Y$. Such an intervention can therefore be considered meaningful, supporting the conclusion that $X$ *is a cause of* $Y$.

Building on this, we formulate two hypotheses grounded in ATE and apply significance tests to assess the causal effects of the textual components $\mathbf{T}$ and the visual components $\mathbf{V}$ on the model's predicted answers $A$.

**Hypothesis 1** *If textual components cause predicted answers? Given visual components $\mathbf{V}$, intervene on textual components $\mathbf{T}$, we have*

$$\begin{cases} H_0^T : ATE^T = 0, \mathbf{T} \text{ does not cause } A \\ H_1^T : ATE^T \neq 0, \mathbf{T} \text{ cause } A \end{cases} \qquad (3)$$

where $ATE^T = E(A|\mathbf{V}, do(\mathbf{T})) - E(A|\mathbf{V}, \mathbf{T})$. Inspired by Lanham et al. (2023), the intervened textual components $do(\mathbf{T})$ are created by injecting mistakes in the original text content with minor modification. We employ GPT-4o (OpenAI, 2024) for mistake injection, with the corresponding prompt detailed in Appendix A.1.1.

**Hypothesis 2** *If visual components cause predicted answers?* *Given textual components* $\mathbf{T}$, *intervene on visual components* $\mathbf{V}$*, we have*

$$\begin{cases} H_0^V : ATE^V = 0, \mathbf{V} \text{ does not cause } A \\ H_1^V : ATE^V \neq 0, \mathbf{V} \text{ cause } A \end{cases} \tag{4}$$

where $ATE^V = E(A|\mathbf{T}, do(\mathbf{V})) - E(A|\mathbf{T}, \mathbf{V})$. For the visual component intervention $do(\mathbf{V})$, we replace the cropped images introduced by zoom-in tool call in MCoT with random noise.

**Remark 1** *In the proposed hypothesis formulation, the ATE captures how much the predicted answer changes under intervention. However, token-level comparison of model predictions for each query is computationally complex. To simplify the analysis, we convert the model's predicted answer into an accuracy measure (1 for correct, 0 for incorrect). The ATE is then defined as the difference between the mean accuracy of all queries with and without intervention. The resulting binary variable enables the use of McNemar's test (McNemar, 1947) to evaluate the significance of ATE (Angrist & Imbens, 1995), revealing the underlying causal relationship.*

### 4.2 QUANTIFYING FAITHFULNESS: RELIABILITY AND SUFFICIENCY

The causal analysis through intervention experiments in Table 1 of Sect. 6.2 indicates that the visual information in MCoT has a limited impact on the model's underlying reasoning process (*i.e.*, the reasoning process relies solely on text information), suggesting that the visual information is less faithful. In this section, we propose a quantitative evaluation pipeline of visual faithfulness for further investigation. This pipeline assesses visual faithfulness from two perspectives: *reliability and sufficiency*. For automated evaluation, a third-party LVLM, *i.e.*, GPT-4o (OpenAI, 2024) is involved as a judger.

**Reliability.** It reflects whether the visual components $\mathbf{V}$ of MCoT are reliable for supporting the predicted answer $A$. In other words, reliability directly reflects the causal consistency between $\mathbf{V}$ and $A$. We leverage GPT-4o model as a judger to assess the reliability of visual evidence for the predicted answer, denoted $\mathcal{J}_R(\mathbf{V}, A)$. The model outputs 'Yes' for reliable evidence and 'No' for unreliable evidence. The prompt for reliability assessment is detailed in Appendix A.1.2. It is formally defined as:

$$\text{Rel}(\mathbf{V}, A) = \mathbb{1}\left[\mathcal{J}_R(\mathbf{V}, A) = \text{'Yes'}\right] \tag{5}$$

**Sufficiency.** It evaluates whether the visual components $\mathbf{V}$ of MCoT contain sufficient information to correctly answer the given question. It is a prerequisite for accurate prediction, while also a key indicator of faithful MCoT reasoning that contains no fabricated or omitted information. GPT-4o is employed again for predicting a new answer from only visual components $\mathbf{V}$, denoted $\mathcal{J}_S(\mathbf{V})$, in which the prompt is detailed in Appendix A.1.3. With ground-truth answer $A_{\mathcal{GT}}$, the sufficiency is derived from the accuracy of the new answer, formally defined as:

$$\text{Suf}(\mathbf{V}) = \mathbb{1}\left[\mathcal{J}_S(\mathbf{V}) = A_{\mathcal{GT}}\right] \tag{6}$$

**Remark 2** *Due to the inherent randomness in the judger model, we perform multiple rounds of judgment and aggregate the results across repetitions to enhance the robustness and reliability of the final outcomes. Available aggregation approaches include 1) majority voting and 2) averaging.*

## 5 SUFFICIENT-COMPONENT CAUSE MODEL LEARNING

**Pitfalls of Existing Methods.** Based on the proposed evaluation pipeline, we conduct extensive assessment on current works (Zheng et al., 2025; Su et al., 2025a), as detailed in Sect. 6.2 and 6.3. Our evaluation reveals that existing methods exhibit less faithfulness in the visual information of their MCoT. Specifically, this visual information is oftentimes not reliable and insufficient for correct answers, which might even be unrelated to the model's final predictions. This suggests that the visual information has minimal impact on the underlying reasoning process. *We attribute this unfaithfulness to their RL reward design*, which only incentivizes the presence of interleaved visual cues via zoom-in tool call, while neglecting the correctness and sufficiency of those visual cues. In other words, their RL reward design may encourage arbitrary/random visual information generated by the zoom-in tool. This design flaw makes the reward easily hacked through introducing arbitrarily ineffective visual cues and deriving the final answer based solely on the textual reasoning. This case is very likely to occur in the earlier stages when visual reasoning ability is underdeveloped, which

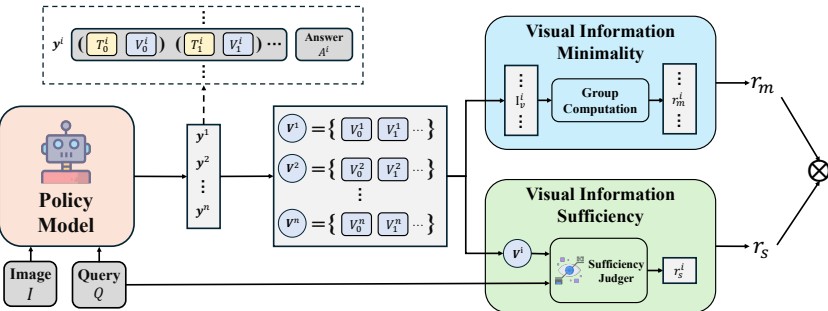

Figure 3: The overview of our proposed Sufficient-Component Cause Model (SCCM) learning to establish visual information as sufficient-component causes to correct answers. The SCCM framework requires that: 1) the visual information alone is sufficient to lead to the correct answer, enforced by the **Visual Information Sufficiency** reward $r_s$; and 2) the visual information involved is as minimal as possible, guided by the **Visual Information Minimality** reward $r_m$.

subsequently makes the visual information largely ignored in the well-trained MCoT, and the model ultimately relies solely on its stronger textual reasoning for the actual reasoning process.

**Toward Mitigating Visual Unfaithfulness.** To improve the accuracy of visual information involved in MCoT and encourage the model to correctly reason with it (*i.e.*, "thinking with images"), we propose the Sufficient-Component Cause Model (SCCM) (Rothman, 1976) learning to establish visual information as a sufficient-component cause for correct answers in RFT training. SCCM required that 1) *Sufficiency*: the visual components in the MCoT must be sufficient to derive the correct answer; and 2) *Minimality*: these sufficient components should be as minimal as possible, without extra irrelevant information (*i.e.*, the zoom-in bounding box should be tightest). The overview of our SCCM learning is illustrated in Figure 3.

**Visual Information Sufficiency.** Our primary objective is to ensure that the visual information $\mathbf{V}$ in MCoT becomes independently sufficient to produce a correct answer. Formally, this requires $\mathrm{Suf}(\mathbf{V}) = \mathbb{1}\left[\mathcal{J}_S(\mathbf{V}) = A_{\mathcal{GT}}\right]$ [1], then the sufficiency reward is:

$$r_{\mathrm{s}}(\mathbf{y}^i) := \mathrm{Suf}(\mathbf{V}^i) = \mathbb{1}\left[\mathcal{J}_S(\mathbf{V}^i) = A_{\mathcal{GT}}\right] \tag{7}$$

where $\mathbf{y}^i$ are the $i$-th rollout of the responses with MCoT, from the same input image $I$ and query $Q$, and $\mathbf{V}^i$ is the visual components extracted from the MCoT of response $\mathbf{y}^i$.

Visual information sufficiency offers several key benefits: (1) it imposes explicit supervision on the visual components, encouraging the model to incorporate accurate and effective visual information; (2) it requires no additional annotations given ground-truth bounding boxes, making it widely applicable and plug-and-play without imposing extra constraints on the training data; (3) by improving the correctness of visual cues, the model becomes more capable of deriving answers from the visual information itself, strengthening causality between visual reasoning and predictions. This avoids unfaithful MCoT which over-relies on textual reasoning and ignores visual reasoning, thereby facilitating better multimodal reasoning.

**Visual information Minimality.** After Sufficiency is achieved, we encourage Minimality to achieve information efficiency. We note that requiring only sufficiency may lead to trivial solutions, *e.g.*, an excessively large region, or even the entire original input image $I$, which serves as a maximally sufficient yet highly inefficient component, with excessive redundant information. Therefore, we introduce an additional *Group Relative Information Minimization* (GRIM) reward during training that favors responses with the tightest bounded visual information within the rollout group. This mechanism encourages the model to leverage minimal sufficient visual information. We illustrate GRIM below:

$$r_m(\mathbf{y}^i) = \frac{\bar{\mathrm{I}}_v}{\mathrm{I}_v(\mathbf{y}^i)}, \quad \bar{\mathrm{I}}_v = \frac{1}{n}\sum_i^n \mathrm{I}_v(\mathbf{y}^i) \tag{8}$$

---

[1]We note that for simplicity, we adopt the definition of $\mathcal{J}_S(\cdot)$ in Eq. 6, while using a more cost-effective model, *i.e.*, Qwen2.5-VL-72B, with simplified prompts for computation efficiency, detailed in Appendix A.2.

where $\mathrm{I}_v(\mathbf{y}^i)$ denotes the total visual information quantity in the MCoT response $\mathbf{y}^i$, which is measured as the total number of image tokens generated from tool calls in MCoT. In other words, we encourage the visual tokens that are shorter than the average visual token throughout $n$ rollouts.

To ensure both sufficiency and minimality simultaneously, the two rewards in Eqs. 7 and 8 are multiplied [2], with a weight value $0 \leq \alpha \leq 1$. Finally, the reward to train faithful MCoT becomes:

$$r_{\text{final}}(\mathbf{y}) = r_{\text{acc}}(\mathbf{y}) + r_{\text{format}}(\mathbf{y}) + \alpha \cdot r_s(\mathbf{y}) \cdot r_m(\mathbf{y}) \tag{9}$$

where $r_{\text{acc}}(\mathbf{y})$ and $r_{\text{format}}(\mathbf{y})$ denote the answer accuracy reward and format reward of response $\mathbf{y}$ that used in the prior arts. The overall reward function is designed to encourage the model to employ MCoT for visual reasoning, ensuring faithfully "thinking with images" that better mimic human behavior (Paivio, 2013).

# 6 EXPERIMENTS

## 6.1 EXPERIMENTAL SETUPS

**MCoT Faithfulness Evaluation Settings.** Our evaluation focuses specifically on Multimodal Chain-of-Thought (MCoT), where visual evidence is explicitly involved in the reasoning process. Accordingly, our evaluation does not consider text-only CoT reasoning, *i.e.*, we exclude cases where the model does not incorporate visual information during reasoning. Tasks that require fine-grained visual perception and understanding naturally emphasize the advantages of MCoT. Therefore, we adopt the *V\* Bench* (Wu & Xie, 2024) that requires the identification of small, query-relevant targets within high-resolution images, to assess such capabilities. HR-Bench (Wang et al., 2025b), which contains images with very high resolutions ranging from 4K to 8K, is also included. Our evaluation includes DeepEyes (Zheng et al., 2025) and Pixel-Reasoner (Su et al., 2025a), both of which integrate visual information into reasoning by the zoom-in tool call, along with visual search method SEAL (Wu & Xie, 2024). We set *Pixel-Reasoner as our primary baseline* for comparison, as our method is built on it with incremental improvements and aligns with its training and reasoning pipelines.

**Training Settings.** Following Su et al. (2025a), we use its publicly released SFT dataset to perform warm-start instruction tuning based on Qwen2.5-VL-7B (Bai et al., 2025), with only image-based question-answering samples included. For SCCM-based RFT, we apply GRPO (Shao et al., 2024) for 80 iterations on $2 \times 8$ A800 (80G) GPUs with the RL training dataset released by Zheng et al. (2025). Each batch contains 128 prompts with 8 rollouts per prompt, allowing a maximum of 6 tool calls per rollout. We configure the KL coefficient to 0.0 and specify the maximum response length as 20480 tokens. More details can be found in Appendix A.2.

## 6.2 RESULTS ON INTERVENTION EXPERIMENTS

We conduct intervention experiments on DeepEyes (Zheng et al., 2025), Pixel-Reasoner (Su et al., 2025a) and our model. As mentioned in Remark 1, the accuracy of model's predicted answer is tested, under (1) No Intervention: the default reasoning process with no intervention; (2) Interv. on $\mathbf{T}$: intervention on the textual components $\mathbf{T}$ of MCoT, for testing Hypothesis 1; and (3) Interv. on $\mathbf{V}$: intervention on the visual components $\mathbf{V}$ of MCoT, for testing Hypothesis 2.

As shown in Table 1 from intervention experiments, DeepEyes (Zheng et al., 2025) and Pixel-Reasoner (Su et al., 2025a) both demonstrate that the visual components have a much weaker causal relation to the predicted answer than the textual components. This suggests that *the visual information involved in MCoT may have less impact on the model's underlying reasoning process*, and the model appears to rely solely on textual reasoning, indicating that MCoT exhibits less faithfulness.

Our model, which incorporates SCCM-based RFT, partially mitigates the issue where visual components have a weak causal relation to predicted answers. It yields a lower $p$-value in testing Hypothesis 2 compared to the baseline models, which means stronger statistical support for the alternative hypothesis $H_1^V$ that visual components cause the predicted answers (Association et al., 2016), suggesting a greater impact of visual information in the MCoT reasoning process.

---

[2]This multiplication reward design prioritizes our primary objective, *i.e.*, sufficiency ($r_s \in \{0, 1\}$), where the minimality reward ($r_m > 0$) contributes as an amplifier for sufficiency, *i.e.*, we obtain positive reward only when sufficiency is satisfied (where $r_s = 1$); otherwise (where $r_s = 0$), we have 0 reward.

(a) Intervention Experiments on DeepEyes.

| Experiments | V* Bench | | | HR-Bench 4K | | | HR-Bench 8K | | |
|---|---|---|---|---|---|---|---|---|---|
| | Attr. | Spat. | Avg. | FSP | FCP | Avg. | FSP | FCP | Avg. |
| No Intervention | 90.43 | 86.84 | 89.00 | 86.75 | 65.75 | 76.25 | 85.50 | 56.00 | 70.75 |
| **Hypothesis 1: If textual components cause predicted answers?** | | | | | | | | | |
| Interv. on **T** | $79.13_{-11.30}$ $0.0023^*$ | $71.05_{-15.79}$ $\mathbf{0.0227}^*$ | $75.92_{-13.09}$ $0.0001^*$ | $71.00_{-15.75}$ $0.0000^*$ | $55.00_{-10.75}$ $0.0000^*$ | $63.00_{-13.25}$ $0.0000^*$ | $75.25_{-10.25}$ $0.0000^*$ | $48.25_{-7.75}$ $0.0039^*$ | $61.75_{-9.00}$ $0.0000^*$ |
| **Hypothesis 2: If visual components cause predicted answers?** | | | | | | | | | |
| Interv. on **V** | $88.69_{-1.74}$ $0.5000$ | $88.16_{+1.32}$ $1.0000$ | $88.48_{-0.52}$ $1.0000$ | $86.50_{-0.25}$ $1.0000$ | $64.75_{-1.00}$ $0.3438$ | $75.62_{-0.63}$ $\mathbf{0.3323}$ | $85.50_{-0.00}$ $1.0000$ | $57.00_{+1.00}$ $\mathbf{0.4240}$ | $71.25_{+0.50}$ $0.5235$ |

(b) Intervention Experiments on Pixel-Reasoner.

| Experiments | V* Bench | | | HR-Bench 4K | | | HR-Bench 8K | | |
|---|---|---|---|---|---|---|---|---|---|
| | Attr. | Spat. | Avg. | FSP | FCP | Avg. | FSP | FCP | Avg. |
| No Intervention | 90.09 | 83.33 | 87.43 | 83.94 | 66.67 | 76.06 | 86.48 | 58.80 | 73.72 |
| **Hypothesis 1: If textual components cause predicted answers?** | | | | | | | | | |
| Interv. on **T** | $78.38_{-11.71}$ $0.0002^*$ | $80.55_{-2.78}$ $0.5000$ | $79.23_{-8.20}$ $0.0001^*$ | $78.50_{-5.44}$ $0.0000^*$ | $63.27_{-3.40}$ $0.0433^*$ | $71.55_{-4.51}$ $0.0000^*$ | $81.89_{-4.59}$ $0.0014^*$ | $57.31_{-1.49}$ $0.4583$ | $70.56_{-3.16}$ $0.0038^*$ |
| **Hypothesis 2: If visual components cause predicted answers?** | | | | | | | | | |
| Interv. on **V** | $90.99_{+0.90}$ $1.0000$ | $83.33_{-0.00}$ $1.0000$ | $87.98_{+0.55}$ $1.0000$ | $84.20_{+0.26}$ $1.0000$ | $66.67_{-0.00}$ $1.0000$ | $76.20_{+0.14}$ $1.0000$ | $84.44_{-2.04}$ $\mathbf{0.0386}^*$ | $59.70_{+0.90}$ $0.6291$ | $73.04_{-0.69}$ $0.4583$ |

(c) Intervention Experiments on our model.

| Experiments | V* Bench | | | HR-Bench 4K | | | HR-Bench 8K | | |
|---|---|---|---|---|---|---|---|---|---|
| | Attr. | Spat. | Avg. | FSP | FCP | Avg. | FSP | FCP | Avg. |
| No Intervention | 93.91 | 86.84 | 91.10 | 85.86 | 58.84 | 72.35 | 89.81 | 54.44 | 72.20 |
| **Hypothesis 1: If textual components cause predicted answers?** | | | | | | | | | |
| Interv. on **T** | $69.56_{-24.35}$ $\mathbf{0.0000}^*$ | $78.95_{-7.89}$ $0.0312^*$ | $73.30_{-17.80}$ $\mathbf{0.0000}^*$ | $62.88_{-22.98}$ $\mathbf{0.0000}^*$ | $46.21_{-12.63}$ $\mathbf{0.0000}^*$ | $54.55_{-17.80}$ $\mathbf{0.0000}^*$ | $70.80_{-19.01}$ $\mathbf{0.0000}^*$ | $39.72_{-14.72}$ $\mathbf{0.0000}^*$ | $55.32_{-16.88}$ $\mathbf{0.0000}^*$ |
| **Hypothesis 2: If visual components cause predicted answers?** | | | | | | | | | |
| Interv. on **V** | $90.43_{-3.48}$ $\mathbf{0.1250}$ | $84.21_{-2.63}$ $\mathbf{0.6250}$ | $87.96_{-3.14}$ $\mathbf{0.0703}$ | $85.35_{-0.51}$ $\mathbf{0.8318}$ | $61.36_{+2.52}$ $\mathbf{0.1325}$ | $73.36_{+1.01}$ $0.3581$ | $88.43_{-1.38}$ $0.2266$ | $53.05_{-1.39}$ $0.4421$ | $70.82_{-1.38}$ $\mathbf{0.1433}$ |

Table 1: We conducted a causal analysis through intervention on the textual and visual components of the generated MCoT from different models on the *V*\* Bench and HR-Bench. The significance of the Average Treatment Effects (ATEs), measured as the difference in mean accuracy, is indicated in red. The corresponding *p*-value for hypothesis testing are shown in blue, where an asterisk (*) denotes statistical significance with a $p$-value $< 0.05$ based on McNemar's test. Lower $p$-value provides a stronger statistical support for the alternative hypothesis, indicating a more pronounced causal effect of the tested components (**T** or **V**) on the predicted answers.

## 6.3 RESULTS ON QUANTITATIVE EVALUATION OF FAITHFULNES

As indicated in Sect. 6.2, the visual information in MCoT exhibits more severe unfaithfulness compared to its textual counterpart. We conduct an extensive evaluation on the faithfulness of the visual components, using the reliability and sufficiency evaluation pipeline introduced in Sect. 4.2.

The evaluation results of reliability and sufficiency for visual components on *V*\* Bench and HR-Bench with different models are shown in Table 2. Our model, incorporating SCCM-based RFT, gains first and second best performance in most tasks, in terms of reliability and sufficiency of the visual components, and shows a significant improvement over the primary baseline, Pixel-Reasoner (Su et al., 2025a), reflecting enhanced faithfulness of visual information. Notably, it outperforms all baseline models on the *V*\* Bench, further demonstrating its stronger capabilities in fine-grained visual perception and understanding. Additionally, our model also achieves superior performance in accuracy, as shown in Table A5 of Appendix A.3.3. The generated MCoT is illustrated in Figure A2 of Appendix A.4 for qualitative and intuitive comparison.

## 7 ABLATION ANALYSIS

To further evaluate the effectiveness of our SCCM learning in RFT, we analyze the training dynamics under different reward schemes: (1) Naive Reward, consisting only of accuracy and format rewards; (2) Curiosity Reward, the curiosity-driven reward scheme following Su et al. (2025a); (3) SCCM Reward, our proposed SCCM reward scheme in Sect. 5; and (4) SCCM w/o Minimality, an ablation variant of SCCM without the minimality constraint. Figure 4 illustrates the training dynamics of RFT from the same warm-start model under these reward schemes on the *V*\* Bench test dataset, including accuracy, visual information sufficiency, the cropped region size for the visual information

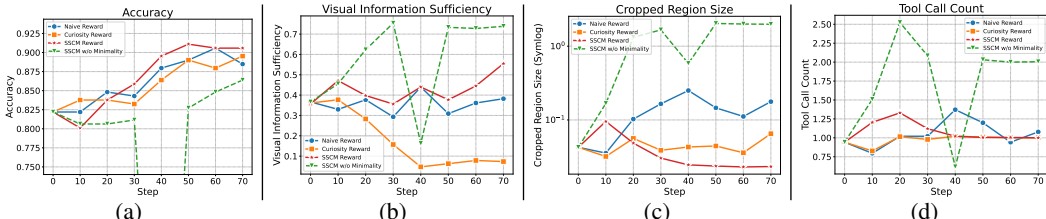

Figure 4: Training dynamics on *V\** Bench as test dataset, with different ablation reward schemes. The visual information sufficiency is judged by Qwen2.5-VL-72B, as detailed in Appendix A.2.

(a) Reliability evaluation results.

| Model | *V\** Bench | | | HR-Bench 4K | | | HR-Bench 8K | | |
|---|---|---|---|---|---|---|---|---|---|
| | **Attr.** | **Spat.** | **Avg.** | **FSP** | **FCP** | **Avg.** | **FSP** | **FCP** | **Avg.** |
| SEAL | 72.73 | 3.95 | 44.62 | 38.46 | 2.56 | 23.08 | 38.33 | 5.00 | 25.00 |
| DeepEyes | 75.73 | 10.45 | 50.00 | **53.00** | 23.25 | **38.12** | 31.56 | 17.34 | 24.45 |
| Pixel-Reasoner | 35.13 | 12.50 | 26.23 | 38.86 | **24.69** | 32.39 | 27.55 | 15.22 | 21.87 |
| Ours | **82.61** | **28.95** | **61.26** | 50.76 | 24.24 | 37.50 | **40.77** | **17.78** | **29.32** |

(b) Sufficiency evaluation results.

| Model | *V\** Bench | | | HR-Bench 4K | | | HR-Bench 8K | | |
|---|---|---|---|---|---|---|---|---|---|
| | **Attr.** | **Spat.** | **Avg.** | **FSP** | **FCP** | **Avg.** | **FSP** | **FCP** | **Avg.** |
| SEAL | 79.09 | 43.42 | 64.52 | 63.46 | 30.77 | 49.45 | 60.00 | 25.00 | **46.00** |
| DeepEyes | 85.44 | 19.40 | 59.41 | 62.50 | 25.75 | 44.12 | 35.35 | 16.33 | 25.84 |
| Pixel-Reasoner | 45.04 | 34.72 | 40.98 | 48.44 | **33.33** | 41.55 | 40.56 | **26.27** | 33.97 |
| Ours | **89.56** | **55.26** | **75.92** | **70.45** | 32.32 | **51.39** | **65.01** | 24.17 | 44.67 |

Table 2: Reliability and sufficiency evaluation results of **visual** components on *V\** Bench and HR-Bench of different models. **Bold** and Underscored denote the first and second best results.

quantity in MCoT and the tool call count. The reliability and sufficiency evaluation results of models under each reward scheme are presented in Table A1 of Appendix A.3.1, and examples of MCoT generated by these models are provided in Figure A3 of Appendix A.4.

**Effectiveness of SCCM in RFT.** As shown in Figure 4b, the proposed SCCM-based reward consistently outperforms both the naive (accuracy and format rewards only) and the curiosity-driven reward in terms of visual sufficiency in MCoT. Notably, the curiosity-driven reward leads to a severe collapse in visual sufficiency, suggesting that simply rewarding the presence of interleaved visual cues without ensuring their correctness can be exploited by the model through ineffective visual cues while disregarding them and still relying primarily on textual reasoning to reach correct answers. Furthermore, as training progresses, the SCCM-based approach also achieves competitive, even superior accuracy, as demonstrated in Figure 4a.

**Crucial Role of the Minimality Constraint.** We investigate the visual information quantity in MCoT by the total aspect ratio of cropped regions relative to the original image, *i.e.*, the cropped region size (Figure 4c). Notably, the absence of the minimality constraint results in excessively large cropped regions, even the entire original input image, and multiple tool calls, which is information inefficient (Figure 4d). While this trivial strategy achieves high visual sufficiency rewards (Figure 4b), it leads to significant training instability. In contrast, the proposed SCCM reward scheme achieves the minimal visual information quantity among all reward schemes and maintains a more stable training process. We also observe that the tool call count under SCCM initially rises before decreasing, eventually stabilizing at 1. This pattern suggests a trial-and-error phase in early training where the model learns to use the zoom-in tool, gradually mastering its effective application.

## 8  CONCLUSION

This paper addresses the critical problem of visual reasoning unfaithfulness in existing MCoT models. We find that while these models appear to generate visual information, which is largely ignored in their Multimodal Chain-of-Thought (MCoT). To diagnose this issue, we first developed a novel evaluation framework to quantitatively analyze the reliability and sufficiency of visual information, revealing that the visual components of existing models are often unreliable, insufficient, and even irrelevant to the final predictions. Building on this analysis, we propose the Sufficient-Component Cause Model (SCCM) learning strategy to enhance the visual faithfulness. Its mechanism requires visual information to serve as a *sufficient and minimal* cause for the correct answer, ensuring the image can independently support the conclusion without redundant details. Our empirical results across multiple benchmarks provide strong evidence that SCCM significantly enhances the faithfulness and accuracy of visual reasoning, offering an effective pathway to ensure "thinking with images" like human beings.

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

# A  APPENDIX

## A.1  DETAILS OF MCOT FAITHFULNESS EVALUATION

### A.1.1  PROMPT FOR MISTAKE INJECTION IN TEXTUAL INTERVENTION

```
**Task Definition**
You are a text modification engine. Strictly follow these rules for any
↪  input:

**Input Structure**
- You will receive:
  1. A `Question` with multiple-choice options
  2. The `Answer` of the question
  3. An `Original sentence` containing reasoning

**Modification Rules**
1. **Single Mistake Requirement**:
   - Introduce exactly ONE mistake related to the question's core
   ↪  subject
   - Choose mistake type based on question content:
     • `Attribute Error`: Modify target object properties
     ↪  (color/size/quantity) to an incorrect value from the options
     • `Relation Error`: Alter spatial/action relationships
     ↪  (position/direction/interaction) to an incorrect state
     ↪  described in the options
     • `Logic Error`: Invert original reasoning conclusions to support
     ↪  an incorrect option
   - The erroneous value MUST be a plausible incorrect option from the
   ↪  Question, distinct from the provided Answer.

2. **Consistency Enforcement**:
   - All references to the modified element MUST be identical
   ↪  throughout the sentence
   - Maintain original wording except for intentional mistake

3. **Context Preservation**:
   - Never alter unrelated details
   - Keep sentence structure identical to original

**Processing Pipeline**
1. Identify the CORE QUESTION SUBJECT (e.g., "color" for color
↪  questions) and the correct value based on the Answer.
2. Select mistake type matching subject:
   - Attribute → Attribute Error
   - Spatial/relational → Relation Error
   - Reasoning-dependent → Logic Error
3. Identify an incorrect target value/state (from the Question's
↪  options) that is plausible and directly contradicts the correct
↪  Answer.
4. Locate ALL instances of the core subject or correct reasoning in the
↪  original sentence.
5. Modify EVERY instance to the SAME erroneous value or conclusion.
6. Verify no other changes exist.

**Critical Failure Prevention**
NEVER allow:
- Contradictions: "The green cart... black color"
- Multiple mistakes: Changing both color and position
- Off-target errors: Modifying unrelated elements
- Value violations: Using values not present in the question options or
↪  using the correct Answer value.
```

```
**Output Format**
ONLY return the modified <think>...</think> sentence with NO
↪   explanations
```

### A.1.2 PROMPT FOR RELIABILITY ASSESSMENT

```
You are an image evidence validator.
Determine if the provided image regions support the entire answer
↪   statement.

**Input:**
- Question: [question about larger image]
- Image regions: One or more crops with bbox [left, top, right, bottom]
- Answer: [textual statement to validate]

**Validation Process:**
1. CAREFULLY EXAMINE THE CONTENT OF ALL PROVIDED IMAGE REGIONS
2. Extract ALL key claims from the Answer
3. For each claim:
    - Check if it's directly visible in any region
    - Or can be logically inferred from visible elements
4. If ALL claims are supported → "Yes"
5. If ANY claim lacks support → "No"

**Critical Rules:**
- Base decisions ONLY on what is visible in the provided regions
- Never use external knowledge (ignore words like "typically" or
↪   "usually")
- Reject if reasoning requires information beyond what's shown

**Examples:**
[Positive]
Regions show: chopped vegetables
Answer: Food preparation in progress
→ "Yes" (core action visible)

[Negative]
Regions show: clothes on floor
Answer: Person changed clothes for work
→ "No" ("work" and "person" not visible)

**Output:**
"Yes" or "No" (single word only)
```

### A.1.3 PROMPT FOR SUFFICIENCY ASSESSMENT

```
You are a helpful visual assistant.

You will receive one or more image crops from a larger scene, each
↪   accompanied by a 2D bounding box (in the format [left, top, right,
↪   bottom]) denoting its position in the original image.

**Example Input**:
`{"bbox_2d": [10, 20, 100, 200]}`

**Your Instructions**:
1.  **Base Answers Solely on Input**: Use **only** the provided visual
↪   crops and their bounding boxes to answer the user's query. **Do not
↪   use any external knowledge or assumptions**.
2.  **Handle Uncertainty**: If the given visual information is
↪   insufficient to answer the query, respond with "I don't know".
3.  **Format Output**: Always place your final answer within
↪   `\boxed{}`.
```

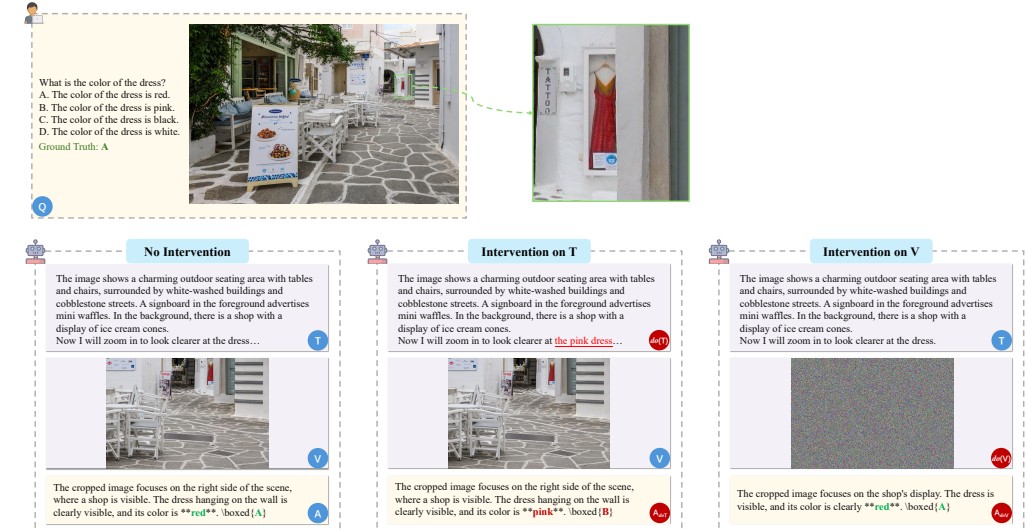

Figure A1: A case from V* Bench showing intervention on MCoT generated by Pixel-Reasoner (Su et al., 2025a). Specifically, the introduced mistake for intervention on textual components is underlined in red. The predicted answer is generated after the MCoT (whether intervened or not).

### A.1.4   A CASE ILLUSTRATION OF INTERVENTION ON MCOT

Figure A1 shows a case of interventions on textual/visual components of MCoT. The final answer is generated under three conditions: (1) No Intervention; (2) Intervention on $\mathbf{T}$, by injecting mistakes into the text; and (3) Intervention on $\mathbf{V}$, by replacing cropped images with random noise. If the final answer changes after an intervention, we identify the intervened component ($\mathbf{T}$ or $\mathbf{V}$) as causal for the prediction of the answer.

## A.2   TRAINING DETAILS

**Instruction Tuning.** We employ the image-based question-answering subset from the publicly available SFT dataset released by Su et al. (2025a). To balance the use of visual operations, we further select 2,700 MCoT trajectories along with 140 text-only trajectories. The model is fine-tuned for one epoch under this configuration using a batch size of 128.

**SCCM-based RFT.** The RL training dataset from DeepEyes (Zheng et al., 2025) is employed as training data, which is carefully curated and thereby facilitates the learning of the zoom-in tool call. We adopt a nearly on-policy paradigm in which the improvement policy is trained with a batch size of 256. We set the coefficient $\alpha = 0.5$ and, to ensure reward numerical stability, clip the group relative visual information reward $r_m(\mathbf{y})$ to the range $[0, 2]$. The visual information sufficiency reward $r_s(\mathbf{y})$ in RFT training is evaluated using Qwen2.5-VL-72B (Bai et al., 2025). The system prompt used to assess visual information sufficiency is provided below.

```
You are a helpful assistant.

You will be provided with one or more input images. Each image is a
↪   cropped section of a larger image, accompanied by its 2D bounding
↪   box. The bounding box is formatted as [left, top, right, bottom].

**Example Input**:
```
{"bbox_2d": [0.1, 0.2, 0.3, 0.4]}
```
```

To further guide the evaluation model, the following prompt is appended after each training query.

```
\n\nGuidelines: Understand the given visual information and the user
↪  query. Answer the user's query based on the given images and
↪  bounding boxes as needed. Please answer as briefly as possible. If
↪  you don't have enough visual information, just answer 'I don't
↪  know'. Always put your final answer within \\boxed{}.
```

**Prompts for Zoom-in Tool.** Following Su et al. (2025a), the zoom-in tool takes a two-dimensional bounding box `bbox_2d` and a `target_image` index that specifies which image to operate on (indexed from 1, with 1 denoting the original image). The system prompt for the zoom-in tool is provided below.

```
You are a helpful assistant.

# Tools

You may call one or more functions to assist with the user query.

You are provided with function signatures within <tools></tools> XML
↪  tags:
<tools>
{"type": "function", "function": {"name": "crop_image", "description":
↪  "Zoom in on the image based on the bounding box coordinates. It is
↪  useful when the object or text in the image is too small to be
↪  seen.", "parameters": {"type": "object", "properties": {"bbox_2d":
↪  {"type": "array", "description": "The bounding box of the region to
↪  zoom in, as [x1, y1, x2, y2], where (x1, y1) is the top-left corner
↪  and (x2, y2) is the bottom-right corner.", "items": {"type":
↪  "number"}}, "target_image": {"type": "number", "description": "The
↪  index of the image to crop. Index from 1 to the number of images.
↪  Choose 1 to operate on original image."}}, "required": ["bbox_2d",
↪  "target_image"]}}}
</tools>

For each function call, return a json object with function name and
↪  arguments within <tool_call></tool_call> XML tags:
<tool_call>
{"name": <function-name>, "arguments": <args-json-object>}
</tool_call>
```

Following Su et al. (2025a), we also append the following prompt after the user query:

```
\nGuidelines: Understand the given visual information and the user
↪  query. Determine if it is beneficial to employ the given visual
↪  operations (tools). For an image, we can look closer by
↪  `crop_image`. Reason with the visual information step by step, and
↪  put your final answer within \\boxed{}.
```

(a) Reliability evaluation results.

| Reward Scheme | V* Bench | | | HR-Bench 4K | | | HR-Bench 8K | | |
|---|---|---|---|---|---|---|---|---|---|
| | **Attr.** | **Spat.** | **Avg.** | **FSP** | **FCP** | **Avg.** | **FSP** | **FCP** | **Avg.** |
| Naive | 27.62 | 7.69 | 20.00 | 27.24 | 19.72 | 23.63 | 16.04 | 7.97 | 12.19 |
| Curiosity | 6.14 | 0.00 | 3.70 | 18.29 | 15.32 | 16.84 | 13.60 | 7.85 | 10.78 |
| SCCM | **82.61** | **28.95** | **61.26** | **50.76** | **24.24** | **37.50** | **40.77** | **17.78** | **29.32** |
| w/o * Minimality | 30.43 | 18.42 | 25.65 | 37.25 | 44.25 | 40.75 | 40.75 | 27.50 | 40.00 |

(b) Sufficiency evaluation results.

| Reward Scheme | V* Bench | | | HR-Bench 4K | | | HR-Bench 8K | | |
|---|---|---|---|---|---|---|---|---|---|
| | **Attr.** | **Spat.** | **Avg.** | **FSP** | **FCP** | **Avg.** | **FSP** | **FCP** | **Avg.** |
| Naive | 31.43 | 24.61 | 28.82 | 39.41 | 30.70 | 35.16 | 30.07 | 23.35 | 26.87 |
| Curiosity | 10.53 | 4.00 | 7.94 | 31.58 | 22.34 | 27.04 | 26.45 | 16.23 | 21.44 |
| SCCM | **89.56** | **55.26** | **75.92** | **70.45** | **32.32** | **51.39** | **65.01** | **24.17** | **44.67** |
| w/o * Minimality | 59.13 | 56.58 | 58.11 | 56.00 | 61.25 | 58.63 | 49.25 | 58.00 | 53.63 |

Table A1: Reliability and sufficiency evaluation results of visual components on V* Bench and HR-Bench of the ablation models under different reward schemes: (1) Naive, consisting only of accuracy and format rewards; (2) Curiosity, the curiosity-driven reward scheme proposed in Su et al. (2025a); (3) SCCM, our proposed SCCM scheme with visual information sufficiency and minimality constraint; (4) SCCM w/o Minimality, an ablation variant of SCCM without the minimality constraint. We note that *SCCM w/o Minimality* (denoted *), is not comparable, as its cropped image is excessively large, being the same size as the original input image.

| Reward Scheme | V* Bench | | HR-Bench 4K | | HR-Bench 8K | |
|---|---|---|---|---|---|---|
| | **CRZ** | **TCC** | **CRZ** | **TCC** | **CRZ** | **TCC** |
| Naive | 0.1490 | 1.4345 | 0.2176 | 1.4587 | 0.1680 | 1.1862 |
| Curiosity | 0.0835 | 0.9895 | 0.1977 | 0.9850 | 0.2144 | 0.9750 |
| SCCM | 0.0429 | 1.0000 | 0.1429 | 1.0000 | 0.1273 | 0.9050 |
| w/o * Minimality | 1.9916 | 2.0000 | 1.9983 | 2.0000 | 1.9682 | 2.0025 |

Table A2: The Cropped Region Size (CRZ), *i.e.*, the total aspect ratio of cropped regions relative to the original image, and Tool Call Count (TCC) on V* Bench and HR-Bench of the ablation models under different reward schemes.

## A.3 ADDITIONAL EXPERIMENTAL RESULTS

### A.3.1 RESULTS OF ABLATION MODELS UNDER DIFFERENT REWARD SCHEMES

We evaluate the reliability and sufficiency of visual components on V* Bench (Wu & Xie, 2024) and HR-Bench (Wang et al., 2025b), with the ablation models under different reward schemes in Sect. 7, illustrated in Table A1. We also show the Cropped Region Size (CRZ), *i.e.*, the total aspect ratio of cropped regions relative to the original image for assessing the visual information quantity in MCoT, and Tool Call Count (TCC) in Table A2. Our proposed SCCM Reward scheme achieves consistent outperformance over the Naive and Curiosity Reward in terms of reliability and sufficiency metrics, further demonstrating the superiority of the proposed SCCM scheme. The model under SCCM without Minimality Reward scheme is not comparable, due to its cropped image being excessively large, *i.e.*, the same size as the original input image with multiple tool calls ($\approx 2$).

(a) Reliability evaluation results.

| RFT Dataset | V* Bench | | | HR-Bench 4K | | | HR-Bench 8K | | |
|---|---|---|---|---|---|---|---|---|---|
| | Attr. | Spat. | Avg. | FSP | FCP | Avg. | FSP | FCP | Avg. |
| Zheng et al. (2025) Dataset (**Ours**) | **82.61** | 28.95 | **61.26** | 50.76 | 24.24 | 37.50 | 40.77 | 17.78 | 29.32 |
| Su et al. (2025a) Dataset (**Ablation**) | 78.07 | **30.67** | 59.26 | **56.82** | **27.29** | **42.13** | **44.00** | **19.49** | **31.82** |

(b) Sufficiency evaluation results.

| RFT Dataset | V* Bench | | | HR-Bench 4K | | | HR-Bench 8K | | |
|---|---|---|---|---|---|---|---|---|---|
| | Attr. | Spat. | Avg. | FSP | FCP | Avg. | FSP | FCP | Avg. |
| Zheng et al. (2025) Dataset (**Ours**) | **89.56** | 55.26 | **75.92** | 70.45 | 32.32 | 51.39 | **65.01** | 24.17 | **44.67** |
| Su et al. (2025a) Dataset (**Ablation**) | 87.72 | **56.00** | 75.13 | **73.74** | **39.79** | **56.85** | 57.25 | **28.10** | 42.77 |

Table A3: Reliability and sufficiency evaluation results of visual components on V* Bench and HR-Bench for models trained via RFT: (1) on the training dataset of Zheng et al. (2025) (*i.e.*, **Ours**), and (2) on the training dataset of Su et al. (2025a) (*i.e.*, **Ablation**).

| Reward Scheme | V* Bench | | HR-Bench 4K | | HR-Bench 8K | |
|---|---|---|---|---|---|---|
| | **CRZ** | **TCC** | **CRZ** | **TCC** | **CRZ** | **TCC** |
| Zheng et al. (2025) Dataset (**Ours**) | 0.0429 | 1.0000 | 0.1429 | 1.0000 | 0.1273 | 0.9050 |
| Su et al. (2025a) Dataset (**Ablation**) | 0.442 | 0.9895 | 0.1584 | 0.9900 | 0.1553 | 0.9950 |

Table A4: The Cropped Region Size (CRZ), *i.e.*, the total aspect ratio of cropped regions relative to the original image, and Tool Call Count (TCC) on V* Bench and HR-Bench of the ablation models under different RFT training datasets.

### A.3.2 RESULTS OF ABLATION MODELS UNDER DIFFERENT TRAINING DATASETS

We also execute RFT using an alternative dataset, *i.e.*, the training dataset of Pixel-Reasoner (Su et al., 2025a), and evaluate the resulting model in terms of reliability and sufficiency. The evaluation results are presented in Table A3, with the cropped region size and tool call count detailed in Table A4. As shown in Table A4, the cropped region size and tool call count of different training dataset settings are similar, suggesting that a comparable visual information quantity is incorporated in their MCoT reasoning processes. The reliability and sufficiency results in Table A3 indicate an overall comparable performance between the two models. However, on HR-Bench, the model trained on the dataset from Su et al. (2025a) (**Ablation**) outperforms the model trained on the dataset from Zheng et al. (2025) (**Ours**), especially on HR-Bench 4K. This difference may be attributed to the training data from Su et al. (2025a), which includes a substantial number of high-resolution images, *e.g.*, from SA-1B (Kirillov et al., 2023). This type of data is therefore likely to enhance the model's ability to perceive and interpret high-resolution visual content.

### A.3.3 RESULTS OF ACCURACY COMPARISON ACROSS DIFFERENT MODELS

Table A5 reports the accuracy results of SEAL (Wu & Xie, 2024), DeepEye (Zheng et al., 2025), Pixel-Reasoner (Su et al., 2025a), and our SCCM-based RFT model on V* Bench and HR-Bench. The evaluation covers all data samples from these benchmarks, including both MCoT and non-MCoT reasoning cases. Our model achieves state-of-the-art performance on the majority of evaluated tasks, particularly on the V* Bench and HR-Bench 8K. Furthermore, it demonstrates a clear and significant improvement over our primary baseline, *i.e.*, Pixel-Reasoner.

| Model | V* Bench | | | HR-Bench 4K | | | HR-Bench 8K | | |
|---|---|---|---|---|---|---|---|---|---|
| | Attr. | Spat. | Avg. | FSP | FCP | Avg. | FSP | FCP | Avg. |
| SEAL | 73.04 | 75.00 | 73.82 | 40.00 | 28.00 | 34.00 | 42.00 | 31.00 | 36.50 |
| DeepEyes | 90.43 | 86.84 | 89.00 | **86.75** | **65.75** | **76.25** | 85.50 | 56.00 | 70.75 |
| Pixel-Reasoner | 88.69 | 81.58 | 85.86 | 83.50 | 60.00 | 71.75 | 86.25 | 53.50 | 69.87 |
| Ours | **93.91** | **86.84** | **91.10** | 86.00 | 59.00 | 72.50 | **86.50** | **56.00** | **71.25** |

Table A5: The accuracy results of different models on *V* Bench and HR-Bench. **Bold** and Underscored denote the first and second best results.

| Model | V* Bench | | HR-Bench 4K | | HR-Bench 8K | |
|---|---|---|---|---|---|---|
| | CRZ | TCC | CRZ | TCC | CRZ | TCC |
| DeepEyes | 0.0074 | 0.9581 | 0.0371 | 1.0250 | 0.0256 | 0.9987 |
| Pixel-Reasoner | 0.0988 | 1.0000 | 0.1076 | 0.9000 | 0.0928 | 0.9275 |
| Ours | 0.0429 | 1.0000 | 0.1429 | 1.0000 | 0.1273 | 0.9050 |

Table A6: The Cropped Region Size (CRZ), *i.e.*, the total aspect ratio of cropped regions relative to the original image, and Tool Call Count (TCC) on *V* Bench and HR-Bench of DeepEyes (Zheng et al., 2025), Pixel-Reasoner (Su et al., 2025a) and our model from SCCM-based RFT.

### A.3.4 STATISTICS ON VISUAL INFORMATION QUANTITY IN MCOT BY DIFFERENT MODELS

We report the statistics on the visual information quantity in MCoT through the cropped region size, *i.e.*, the total aspect ratio of cropped regions relative to the original image, and the zoom-in tool call count is also included. Results are provided for DeepEye (Zheng et al., 2025), Pixel-Reasoner (Su et al., 2025a), and our SCCM-based RFT model, as shown in Table A6.

Combining the results in Table 2, it suggests that DeepEye incorporates extremely small cropped regions, which often provide insufficient visual information. This likely explains its suboptimal performance in terms of reliability and sufficiency, especially in scenarios involving multiple target objects or large objects, *e.g.*, queries from HR-Bench. In comparison, Pixel-Reasoner crops larger regions, but these often include substantial query-unrelated visual content. In contrast, our model, using SCCM, maintains visual cues of an appropriate size, *i.e.*, with suitable information quantity, while ensuring both the correctness and effectiveness of the visual information.

### A.4 MORE CASES

• Comparison of the Generated MCoT by Different Models

We provide some cases in *V* Bench with responses generated by DeepEyes (Zheng et al., 2025), Pixel-Reasoner (Su et al., 2025a) and our SCCM-based RFT model, illustrated in Figure A2. We observe that both DeepEyes and Pixel-Reasoner incorporate incorrect visual evidence during their MCoT reasoning, whereas our model integrates accurate visual cues and exhibits a more rational reasoning process.

• Comparison of the Generated MCoT by Models in Different Reward Schemes

Cases in *V* Bench with responses generated by models in different reward schemes are shown in Figure A3. Models under curiosity reward and naive reward both exhibit incorrect visual cues, and yield predicted answers that disregard these cues, indicating that the absence of supervision on the involved visual information in MCoT can easily lead to the problems of inaccurate visual cues and unfaithful reasoning processes. In contrast, the model trained with SCCM reward demonstrates more accurate visual cues and a more rational reasoning process. However, without the minimality constraint, it incorporates the original input image, which is excessively large, even twice in the MCoT. This suggests that the image sufficiency reward alone can be easily exploited, underscoring the necessity of the minimality constraint in the SCCM scheme.

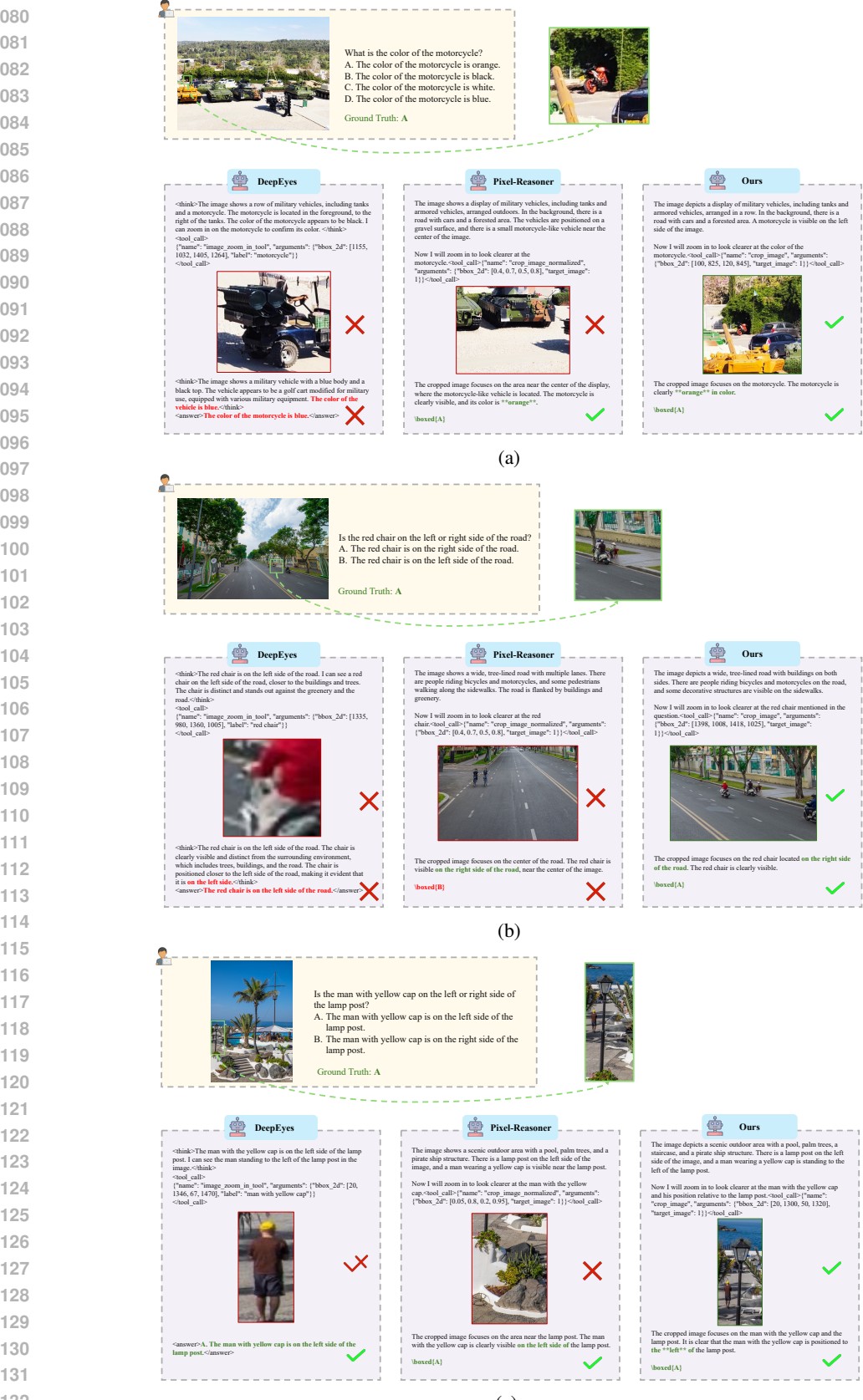

Figure A2: Cases in *V\** Bench with MCoT responses generated by DeepEyes (Zheng et al., 2025), Pixel-Reasoner (Su et al., 2025a) and our SCCM-based RFT model.

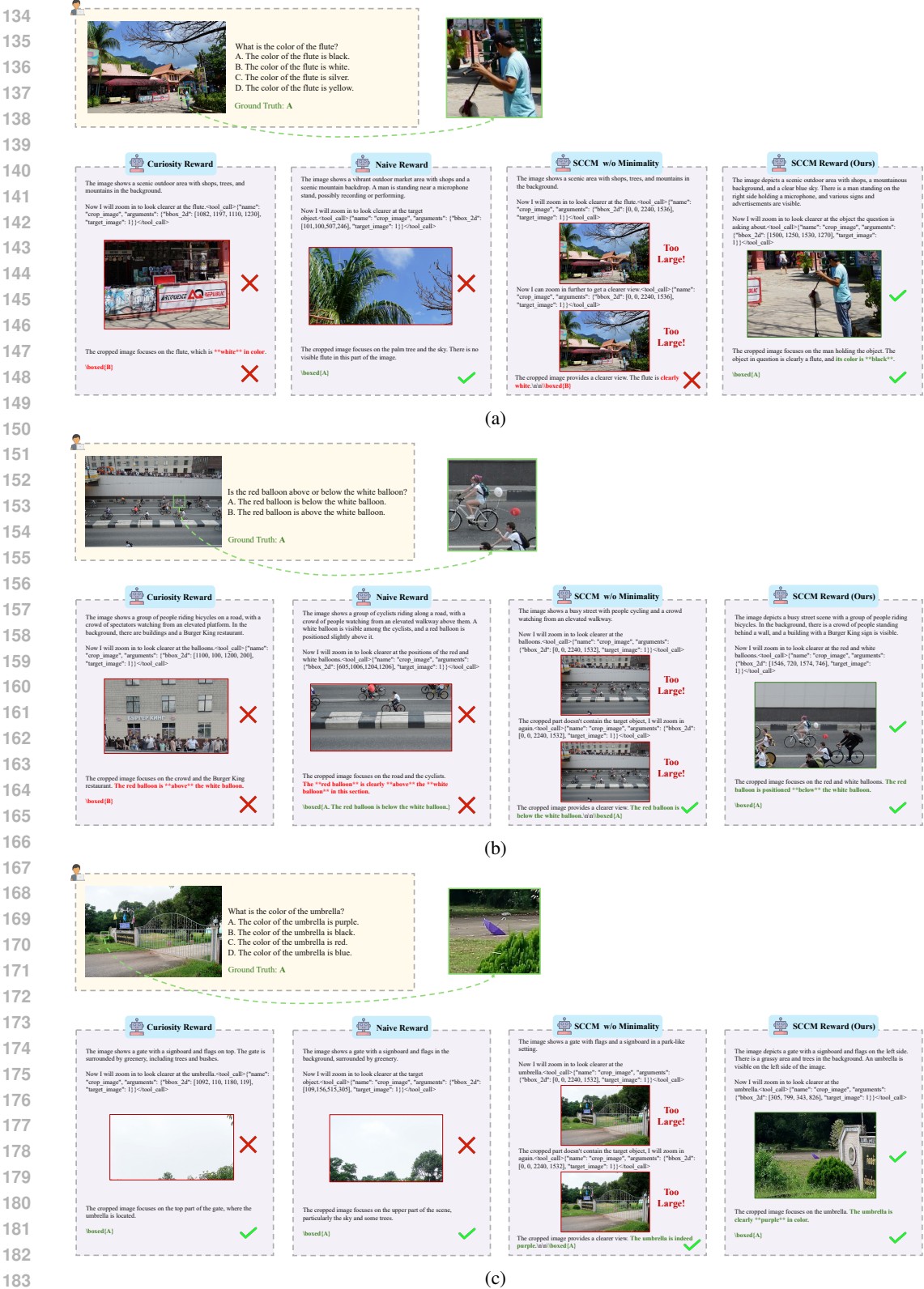

Figure A3: Cases in *V\** Bench with MCoT responses generated by models in different reward schemes: (1) Curiosity Reward, the curiosity-driven reward scheme proposed in Su et al. (2025a); (2) Naive Reward, consisting only of accuracy and format rewards; (3) SCCM w/o Minimality, an ablation variant of SCCM without the minimality constraint; (4) SCCM Reward, our proposed SCCM scheme with visual information sufficiency and minimality constraint.

