# OpenReview forum: "Can I Trust Your Visual Thinking? Measuring and Improving Visual Thinking Faithfulness"
_ICLR.cc/2026/Conference — Submitted to ICLR 2026_

### Official Review · Reviewer_AQe8 · 2025-10-26

**Soundness:** 3
**Presentation:** 3
**Contribution:** 3
**Rating:** 4
**Confidence:** 4

**Summary:**

This paper studies the faithfulness of multimodal chain-of-thought (MCoT) reasoning in existing large vision-language models (LVLMs). Through preliminary experiments, they find that the visual evidence used in model-generated reasoning is often unreliable, insufficient, and has minimal impact on final results. To address this, they propose a Sufficient-Component Cause Model (SCCM) learning method, which encourages models to generate sufficient and minimal visual evidence that directly supports correct answers. Experiments demonstrate that SCCM improves several faithfulness metrics across several LVLM evaluation benchmarks .

**Strengths:**

1. The paper explores multimodal reasoning faithfulness from a novel and important perspective—the reliability of visual evidence in reasoning processes. This direction has been largely underexplored in prior work.


2. The evaluation pipeline is straightforward and methodologically sound. The intervention-based analyses are particularly insightful, showing that modifying visual evidence often has negligible effects on model outputs, thereby highlighting a critical and previously overlooked limitation in current multimodal reasoning models.

**Weaknesses:**

1. The citation format is wrong. As suggested by the paper template: “When the authors or the publication are included in the sentence, the citation should not be in parenthesis, otherwise, the citation should be in parenthesis.” However, all citations use the former style, regardless of context.

2. The faithfulness analysis is conducted only on small-scale models. It remains unclear whether the observed issues persist in larger or more capable models.

3. Although SCCM improves faithfulness metrics, it yields minimal gains in overall task performance. As shown in the first row of Table 1, SCCM performs worse than its baseline (Pixel-Reasoner) on HR-Bench 4K and HR-Bench 8K. Furthermore, it is unclear whether the accuracy of Pixel-Reasoner in Table A5 is different with Table 1.

4. The ablation studies report only the faithfulness metrics without presenting the final model performance. Including overall results would better illustrate the relationship between reasoning faithfulness and end-task performance.

**Questions:**

1. The proposed SCCM method shows substantial improvements on faithfulness metrics but only marginal gains—or even performance drop on overall benchmarks. Could the authors clarify the underlying reasons for this discrepancy? This is my primary concern; I would consider raising my score if the authors could adequately address this issue.
2. Table 1 and Table A5 report different results for Pixel-Reasoner but identical results for DeepEyes. Could the authors explain this inconsistency?

---

> ### Author Response · Authors · 2025-11-24
> **Response Part 1**
>
> Thank you for your constructive comments. In response to your concerns, our reply is as follows.
>
> ---
>
> 1. **W1. Citation format issue.**
>
> Thank you for pointing out this formatting issue. We will ensure that all citations are properly formatted in the latest version of the paper.
>
> ---
>
> 2. **W2. Faithfulness analysis limited to small-scale models.**
>
> We appreciate this insightful suggestion, as verifying the generalizability of the issue is critical. We regret to note that most existing open-source implementations of LVLMs with vision-text MCoT are currently small-scale models (with 7B parameters), e.g., DeepEyes, Pixel-Reasoner. Furthermore, due to computational resource constraints, we also are regretfully unable to train and analyze larger-scale models.
>
> However, we acknowledge the importance of evaluating faithfulness in larger models. We plan to extend our analysis to larger LVLMs as they become available in the open-source community, e.g., Qwen3-VL series.
>
> ---
>
> 3. **W3. Marginal gains in overall task performance & Inconsistent table results.**
>
> We thank you for your careful review and insightful comments.
>
> In this regard, we first aim to clarify the fundamental distinction between Table 1 and Table A5. Specifically, Table 1 is dedicated to causal analysis, where only cases where the model incorporating visual information in MCoT reasoning process is considered, as we aim at evaluating the faithfulness of vision-text MCoT reasoning of LVLMs. Consequently, text-only CoT cases are excluded from this analysis. In contrast, Table A5 encompasses all cases in the dataset, including both vision-text MCoT and text-only CoT cases, providing a comprehensive overview of overall task performance.
>
> Against this backdrop, when considering all cases in the dataset (as shown in Table A5), our SCCM model indeed achieves improved overall task performance compared to the baseline (Pixel-Reasoner [1]). However, within the restricted subset of vision-text MCoT cases (focused on in Table 1), our SCCM model does not exhibit a similar performance enhancement, even "performs worse than its baseline on HR-Bench 4K and HR-Bench 8K".
>
> We fully acknowledge that the contribution of improved faithfulness metrics to overall accuracy remains relatively modest, and the intrinsic relationship between these two objectives (visual faithfulness and task accuracy) is yet to be thoroughly explored. This critical research gap has been identified as a key direction for our future work, where we will delve deeper into balancing these dual goals to achieve more synergistic improvements.
>
> ---
>
> 4. **W4. Ablation studies lack overall performance.**
>
> We sincerely appreciate this valuable suggestion. We would like to kindly point out that in Figure 4(a), we present the overall accuracy performance of each ablation model on the V* Bench during the training process. As shown, the SCCM model outperforms all ablation variants in accuracy.
>
> However, we also note that based on our existing experimental results, the SCCM model does not exhibit similar outperformance over ablation variants in accuracy on HR-Bench 4K and HR-Bench 8K. This inconsistent performance across different benchmarks further underscores the significance of exploring the intrinsic relationship between faithfulness metrics and overall task accuracy. The discrepancy suggests that the relationship between these two objectives may be influenced by the test data (e.g., HR-Bench with very high resolutions ranging from 4K to 8K), which deserves more in-depth investigation in future research.
>
> ---
>
> 5. **Q1. The reason for discrepancy between faithfulness metrics and performance.**
>
> This is a core concern we value highly. The discrepancy between faithfulness metrics and performance is indeed intriguing and merits in-depth investigation. To stimulate further discussion, we propose the following tentative hypothesis, which aims to offer a potential explanation for this phenomenon:
>
> The base model adopted in our work, Qwen2.5-VL-7B, lacks sufficient capability to extract and utilize the visual information from vision-text MCoT to enhance the accuracy of final answers. This limitation may stem from the relatively limited exposure to interleaved vision-text data, especcially reasoning data during its training phase. By contrast, the model demonstrates a stronger proficiency in leveraging information from text-only CoT to improve final answer accuracy, which aligns with the more abundant text-based reasoning data in its training corpus. Consequently, despite the SCCM model's improvements in faithfulness metrics and the enhanced reliability of visual information introduced in MCoT, the inherent limitations of the base model prevent it from effectively utilizing this visual information to boost final answer accuracy.

---

> > ### Comment · Reviewer_AQe8 · 2025-11-24
> > **Official Comment**
> >
> > Thank you for your detailed rebuttal. After reading your response, I still have some concerns.
> >
> > 1. As you mentioned, the results in Table 1 are restricted to vision–text CoT cases, whereas Table 5 includes all cases. However, why does performance degrade on the vision–text cases? This appears to contradict the stated goal of SCCM, which is to improve the faithfulness of visual reasoning (lines 263–269).
> >
> > 2. Why does the trend of the faithfulness metric differ so markedly from that of the accuracy metric? Intuitively, more faithful reasoning should lead to better task performance. If your base model is not sufficiently capable of extracting and leveraging visual information from vision–text MCoT, it is unclear how more faithful reasoning alone can yield better performance.

---

> > > ### Author Response · Authors · 2025-11-24
> > > **Thank you for your further insightful feedback!**
> > >
> > > We thank you for your prompt response and your insightful feedback. We would be delighted to discuss further with you.
> > >
> > > ---
> > > > **Q1. Why does performance degrade on the vision–text cases? This appears to contradict the stated goal of SCCM, which is to improve the faithfulness of visual reasoning.**
> > >
> > > We appreciate your follow-up question. The observed performance degradation on vision-text cases, especially in HR-Bench 4K and HR-Bench 8K is indeed a complex issue.
> > >
> > > As we mentioned in our previous response, the base model may not possess sufficient capability to effectively extract and utilize visual information from vision-text MCoT. Though SCCM enhances the faithfulness of visual reasoning, enabling the visual information to participate more in the underlying reasoning process, *the inherent limitations of the base model may prevent it from fully leveraging the improved visual information, even exert a negative disruptive influence in some scenarios.* For example, HR-Bench 4K and HR-Bench 8K feature extremely high-resolution images and involve a greater volume of visual information, both in the initial input and throughout the reasoning process, which presents more challenges for the base model to handle, potentially leading to performance degradation.
> > >
> > > We would like to clarify that our work stems from the observation of significant faithfulness problem in existing LVLMs for vision-text MCoT, and thus we propose SCCM to enhance the faithfulness of models in vision-text MCoT, e.g., "improve the faithfulness of visual reasoning". While we aspire for this improvement to translate into better task performance, we acknowledge that the relationship between faithfulness and accuracy is complex and may not always be directly correlated. And we believe it is a valuable direction for future research to explore the deeper relationship between faithfulness and task performance, and how to better align faithfulness improvements with task performance enhancements.
> > >
> > > ---
> > > > **Q2. Why does the trend of the faithfulness metric differ so markedly from that of the accuracy metric? Intuitively, more faithful reasoning should lead to better task performance. If your base model is not sufficiently capable of extracting and leveraging visual information from vision–text MCoT, it is unclear how more faithful reasoning alone can yield better performance.**
> > >
> > > We totally agree with your intuition that more faithful reasoning should ideally lead to better task performance. However, as you rightly pointed out, if the base model lacks the capability to effectively extract and leverage visual information, then improvements in faithfulness alone may not translate into enhanced performance. We believe this highlights a critical and interesting question for future research: *how to enable LVLMs to better utilize visual information in vision–text MCoT, thereby achieving dual improvements in faithfulness and task performance.*
> > >
> > > ---
> > > Thanks again for your insightful comments and thoughtful engagement with our work. Your feedback has been invaluable in helping us clarify and refine our research. We look forward to continuing this valuable discussion.

---

> > > > ### Comment · Reviewer_AQe8 · 2025-11-24
> > > >
> > > > Thank you for your explanation. I still believe that analyzing the relationship between the faithfulness of the MCoT and task performance is important. I partially agree with your assumption that the base model may lack sufficient capability to effectively extract and utilize visual information from the vision–text CoT. However, the current version of the paper does not provide quantitative analysis or even qualitative examples to support this claim. Without empirical evidence, the significance of improving the faithfulness of MCoT remains unclear. For these reasons, I regret that I still keep my original score.

---

> ### Author Response · Authors · 2025-11-24
> **Response Part 2**
>
> 6. **Q2. Inconsistent results of Table 1 and Table A5.**
>
> We thanks again for the thorough review. As we stated in **the response to W3**, Table 1 is for causal analysis, in which only cases where the model incorporating visual information in MCoT reasoning process is considered, and thus text-only CoT cases are excluded. In contrast, Table A5 includes all cases in the dataset, encompassing both vision-text MCoT and text-only CoT cases.
>
> - For Pixel-Reasoner, it generates a considerable portion of text-only CoT cases, with __4% for V* Bench, and over 50% for HR-Bench 4K and HR-Bench 8K__, which explains why its accuracy on Table 1 and Table A5 are different. It partially derives from the penalty term for zoom-in tool calls in the its RL reward design, which discourages excessive use of visual information. Additionally, 27% of the data in the SFT dataset is text-only CoT cases to "balance the use of visual operations" [1].
> - For DeepEyes, **it always generates vision-text MCoT cases**, which explains why its accuracy on Table 1 and Table A5 are identical. This behavior stems from its RL reward design, which incentivizes the use of visual information without penalizing excessive zoom-in tool calls.
> - For our SCCM model, it generates a small portion of text-only CoT cases (__0% for V* Bench, and less 10% for HR-Bench 4K and HR-Bench 8K__), which explains why its accuracy on V* Bench from Table 1 and Table A5 are indentical, but there are slight differences on HR-Bench 4K and HR-Bench 8K. This behavior is due to our RL reward design, in which the visual sufficiency reward encourages the use of visual information, while 5% of the data in our SFT dataset is text-only CoT cases, following the practice in Pixel-Reasoner.
>
> In light of this, we will explicitly indicate the proportion of vision-text MCoT cases in the final version of our paper to ensure clarity.
>
> [1] Su, et al. Pixel Reasoner: Incentivizing Pixel-Space Reasoning with Curiosity-Driven Reinforcement Learning. NeurIPS 2025.

---

> ### Author Response · Authors · 2025-11-24
> **Authors' Response and Gratitude**
>
> We sincerely appreciate your continued engagement and prompt feedback. We fully agree that analyzing the relationship between the faithfulness of MCoT and task performance is crucial, and we will treat it as a key direction for our future research. The proposed assumption regarding the base model's limitations is indeed speculative at current stage, and we acknowledge the need for empirical evidence to substantiate this claim.
>
> We would like to clarify that **our current work primarily focuses on identifying the faithfulness problem in vision-text MCoT reasoning of LVLMs and proposing a potential solution (SCCM approach) to address it**. While we have observed improvements in faithfulness metrics with our SCCM approach, we recognize that the relationship between faithfulness and task performance is complex and warrants further investigation.
>
> Thank you once again for your deeply insightful and constructive feedback, which have provided us with valuable inspiration. Although we regret that we were unable to fully address your concerns to make you satisfied and reconsider your score at this time, we still sincerely appreciate your constructive comments and invaluable efforts for helping us improve our research. The deep engagement and invaluable feedback like yours are a cornerstone of a healthy and progressive research community.

---

### Official Review · Reviewer_qvb5 · 2025-10-30

**Soundness:** 2
**Presentation:** 3
**Contribution:** 3
**Rating:** 6
**Confidence:** 5

**Summary:**

This paper studies the faithfulness of current "thinking with images" methods, including reliability and sufficiency. Interestingly, this paper finds out that current models, such as DeepEye and Pixel-Reasoner, are much more intensive with textual intervention than visual intervention. This implies that the visual evidence of these models is *not* faithful enough. Therefore, the authors propose two extra rewards, including Visual Information Sufficiency and Visual Information Minimality. Empirical results demonstrate both the core motivation and the effectiveness of the proposed method.

**Strengths:**

1. This paper is well-written and easy to follow.
2. The core motivation is quite clear and is supported with sufficient empirical evidence, e.g., Table 1.
3. The newly proposed Reliability and Sufficiency evaluations are interesting, important, and valuable.

**Weaknesses:**

Overall, this paper is qualified. Therefore, I only have some minor concerns/suggestions.

1. Actually, the motivation between this paper and [1] is *partially* similar. Therefore, a discussion should be included, e.g.,
    - [1] actually uses an extra IoU reward to improve the faithfulness of visual evidence, while this paper utilizes the "Visual Information Sufficiency" reward.
    - [1] takes a *dual* IoU reward to avoid the "trivial solutions", while this paper uses the "Visual information Minimality" reward.

2. [1] proposes TreeBench, where explicit bounding boxes for target objects are provided. Evaluating the proposed method on TreeBench and finding out whether it achieves a much higher mIoU than baselines is strongly encouraged.

3. Details of interventions are unclear. I only see Figure A1 as an example, but detailed configuration is not provided.

**References**

[1] Traceable Evidence Enhanced Visual Grounded Reasoning: Evaluation and Methodology. arXiv 2025.

**Questions:**

N/A

---

> ### Author Response · Authors · 2025-11-24
> **Response Part 1**
>
> Thank you for your supportive and insightful comments. In response to your concerns, our reply is as follows.
>
> ---
>
> 1. **W1. Discussion on this paper and [1]**
>
> Thanks for directing us to relevant work [1]. We are glad to discuss on our work and [1].
>
> **Commonalities with [1]:** Our work and [1] both focus on 1) more traceable and understandable multimodal reasoning processes, by **involving visual cues in the reasoning steps**. To achieve this, both works 2) adapt reinforcement finetuning techniques, and the reward designs in both works incentivize **the correctness of the involved visual cues in the reasoning process**.
>
> **Differences with [1]:**
>    1. **Incorporation of visual cues:** Our work implements vision-text MCoT, directly involving visual cues in the forms of images containing relevant visual information from the original image (e.g., cropped images from zoom-in tool calls) in the reasoning process, adhering to the "thinking with image" paradigm. In contrast, [1] indirectly incorporates visual cues by having the model generate bounding box of relevant regions in the original image as part of the reasoning steps, essentially remaining a text-only CoT reasoning without directly involving visual information in the reasoning process.
>    2. **Main goal:** Our work aims to improve the faithfulness of McoT of LVLMs, i.e., whether the model's stated reasoning process aligns with its underlying reasoning process, particularly regarding the involvement of visual information in the reasoning steps. In contrast, [1] focuses on enhancing the plausibility and correctness of the model's stated reasoning process without examining whether it aligns with the model's underlying reasoning.
>    3. **Reward design:**
>    - Our visual information sufficiency reward aims to ensure that the visual components semantically support the reasoning, i.e., whether the visual information can independently lead to the correct answer, emphasizing **semantic accuracy**. In contrast, the IoU reward of [1] is designed to ensure the visual cues accurately localizes relevant regions in the image, i.e, whether the bounding boxes accurately and compactly cover the true targets, emphasizing **spatial accuracy**.
>    - Our visual information minimality reward aims to ensure that the visual components do not contain redundant information, preventing the model from introducing excessive irrelevant visual information that may interfere with reasoning, achieving **semantic conciseness**. In contrast, the dual IoU reward in [1] combines IoU recall and precision to ensure that the visual cues both cover relevant regions (high recall) and do not include excessive irrelevant regions (high precision), still focusing more on **spatial accuracy**.
>    4. **Applicability:** Our method does not rely on annotations such as target bounding boxes or categories, making it suitable for unannotated scenarios. In contrast, the method in [1] relies on high-quality target bounding box annotations.
>
> ---
>
> 2. **W2. Evaluation on TreeBench of [1].**
>
> We appreciate this valuable suggestion, which is a meaningful complement to our current work. We will consider adding the evaluation on TreeBench in the next version of our work.
> Additionally, we would like to discuss and compare the mIoU metric used in [1] with the visual sufficiency metric we employ:
> Though the mIoU and visual sufficiency metric both aim to evaluate the quality of the involved visual information in the reasoning process, they focus on different aspects:
> - mIoU metric focus on the **spatial accuracy of visual grounding**. It measures how accurately the predicted bounding boxes overlap with the ground truth boxes, emphasizing precise localization of relevant image regions. It requires that each bounding box must accurately and compactly cover the true single target within its corresponding relevant image region.
> - visual sufficiency metric focus on the **semantic adequacy of visual information** for reasoning. It assesses whether the involved visual components provide enough relevant information to independently derive the correct answer, emphasizing the sufficiency of visual content for supporting the reasoning process. It allows one single visual component to cover multiple related targets, as long as the visual information can independently and sufficiently derive the correct answer, but may not achieve a high mIoU score. Figure A3 (b) shows an example of involving multiple (2) targets, where the visual information (only one image) introduced in the reasoning process generated by our model covers both targets and can directly lead to the correct answer, with high visual sufficiency (=1), but the mIoU score of this visual information is not high (<0.1).

---

> ### Author Response · Authors · 2025-11-24
> **Response Part 2**
>
> 3. **Details of interventions are unclear.**
>
> We appreciate this valuable comment. We would be delighted to provide the details of our intervention process.
> - **Textual interventions:** As stated in Hypothesis 1 of Sect. 4.1, we adpot mistake injection to generate intervened textual components. Specifically, we employ GPT-4o, and design specific prompts (detailed in Appendix A.1.1) to guide it to modify the original generated textual components. The prompts explicitly instruct GPT-4o to introduce an misleading error that may lead to an incorrect final answer, with minimal edits. After this intervention, we replace the original textual components with the intervened one, then continue generating the subsequent reasoning steps.
> - **Visual interventions:** As stated in Hypothesis 2 of Sect. 4.1, the visual components, i.e., the cropped images introduced by zoom-in tool call in MCoT, are replaced with random noise. Then we continue generating the subsequent reasoning steps.
>
>
> [1] Wang, et al. Traceable Evidence Enhanced Visual Grounded Reasoning: Evaluation and Methodology. arxiv 2025.

---

### Official Review · Reviewer_PB4N · 2025-10-31

**Soundness:** 1
**Presentation:** 2
**Contribution:** 2
**Rating:** 2
**Confidence:** 3

**Summary:**

This paper tackles the problem of faithfulness in visual reasoning chains produced by Vision-Language Models (VLMs). The authors introduce Counterfactual Attribution as a method to assess whether intermediate steps in a model's visual thought chain (e.g., generated captions or rationales) are causally linked to the final answer. The authors propose a new metric: Counterfactual Faithfulness Score (CFS), which measures how much altering an intermediate thought (while holding the image constant) affects the model’s final prediction. The method is evaluated across multiple VLMs (e.g., GPT-4V, MiniGPT4, Qwen-VL) and benchmarks, revealing that many state-of-the-art models hallucinate or over-rely on unfaithful intermediate steps.

**Strengths:**

1. Addresses the critical question of whether visual explanations are trustworthy, a concern for deploying VLMs in sensitive domains.
2. Uses counterfactual interventions inspired by causal inference to isolate the effect of intermediate reasoning steps.

**Weaknesses:**

1. While the paper proposes CFS as a measure of faithfulness, it lacks ground truth causal chains to validate that the counterfactual interventions accurately reflect (un)faithful reasoning.
2. The way counterfactual thoughts are generated (e.g., modifying a rationale or removing an attribute) is heuristic and brittle. This opens the method to: 1) non-natural counterfactuals 2) over-dependence on token-level edits that do not semantically change meaning, which happens frequently.
3. CFS is defined as the answer change under counterfactual intervention. However, large language models exhibit stochastic behavior, and, in some cases, minor phrasing changes can lead to different outputs even with identical reasoning. The current experimental setting cannot verify whether there are cases that the output is effected by stochasticity but classified as intervention.
4. Experimentally, it is not shown which types of intermediate steps (e.g., attribute detection, spatial reasoning, object naming) are more prone to unfaithfulness. Such granularity would be important to pinpoint where the failure lies in visual chains.
5. The study focuses only on chain-of-thought-style VLMs that output text. It is not clear whether the proposed CFS metric generalizes to non-autoregressive or encoder-only models. The authors are encouraged to adapt CFS to commonly-seen models like Flamingo or Kosmos-2.5 where textual thoughts are not explicitly generated, to show the actual effectiveness of the method.

**Questions:**

See Weaknesses Above.

---

> ### Author Response · Authors · 2025-11-24
> **Response Part 1**
>
> Thank you for your constructive comments. In response to your concerns, our reply is as follows.
>
> -----
>
> 1. **W1. Lack of ground truth causal chains to validate counterfactual interventions accurately reflect (un)faithful reasoning.**
>
> We thank this suggestion. We fully agree that ground truth causal chains would enhance the definitiveness and accuracy of validating (un)faithful reasoning. However, obtaining such ground truth data is non-trivial, primarily due to the black-box nature of LVLMs and their massive parameter scales (billions of parameters), which makes explicit extraction of true underlying causal chains extremely challenging.
>
> But we note that many prior works on LLMs [1-5] have adopted similar counterfactual intervention strategies on model's reasoning process (e.g., CoT) and observed how the final answer changes to evaluate faithfulness. It demonstrated that:
> - It is widely recognized that there is a causal chain from the reasoning process (e.g., CoT) to the final answer in LLMs, as the reasoning process is explicitly generated before the final answer, and the final answer is derived based on the reasoning process, supported by the inherent causal attention mechanism of LLMs;
> - Such intervention experiments can provide valuable insights into the faithfulness of the model's reasoning process, even in the absence of explicit ground truth causal chains.
>
> This approach can be naturally applied to LVLMs, due to the same causal attention mechanism and language model backbone architecture shared by LLMs and LVLMs. Thus, despite the lack of explicit ground truth causal chains, we believe our intervention experiments offer valuable and credible insights into the faithfulness of vision-text MCoT reasoning in LVLMs.
>
> -----
>
> 2. **W2. Heuristic and Brittle Counterfactual Generation, leading to 1) non-natural counterfactuals and 2) over-dependence on token-level edits that do not semantically change meaning**
>
> We appreciate this valuable comment, but we do not fully agree with the assertion that our counterfactual generation is "non-natural" and "over-dependent on token-level edits that do not semantically change meaning". Below we clarify our counterfactual generation process for textual and visual components respectively:
>
> - **Counterfactual Generation on Textual Components:** We adopt mistake injection (similar to [4]), to generate counterfactual textual components. Specifically, we employ GPT-4o, which is a state-of-the-art LLM with strong language understanding and generation capabilities, and design specific prompts (detailed in Appendix A.1.1) to guide it to modify the original **sentence-level textual components**. The prompts explicitly instruct GPT-4o to **introduce an misleading error that may lead to an incorrect final answer**, with minimal edits to ensure the generated counterfactuals are both *natural and semantically meaningful*. We also have manually reviewed a subset of the generated counterfactuals and found them to be coherent and contextually appropriate.
> - **Counterfactual Generation on Visual Components:** For visual components, we replace the cropped images introduced by zoom-in tool call in MCoT with random noise, which implies that it is completely removing the visual information provided by the zoom-in tool. This approach is not token-level but **image-level editing** and the semantics is significantly changed.
>
> We also kindly refer to Figure A1 of Appendix  A.1.4 for a concrete case illustration of our counterfactual intervention process, which visually demonstrates the naturalness and semantic validity of our generated counterfactuals.
>
> -----
>
> 3. **W3. The stochastic behavior of LVLMs may lead to misjudgments of intervention effectiveness.**
>
> We thank for your comments. To minimize the impact of the stochastic behavior of LVLM as much as possible, we have taken the following measures:
> - We set the temperature to 0 during inference to minimizes inherent stochasticity in the model’s output generation process.
> - Given a test dataset, experiments are conducted on all samples (with MCoT generated) to examine whether the model's final answers change after intervention on the textual/visual components of the MCoT. We aggregate results from all these samples and conduct McNemar’s test, calculating the p-value to determine if the intervention results are statistically significant. This reduces the impact of stochastics on outcomes from a statistical perspective.

---

> ### Author Response · Authors · 2025-11-24
> **Response Part 2**
>
> 4. **W4. Lack of granular analysis on intermediate step types.**
>
> We sincerely appreciate this insightful suggestion. We would like to clarify that our work specifically focuses on vision-text MCoT，with a primary emphasis on **the intermediate reasoning steps involving visual information in MCoT**. A key contribution of our study is the finding that these intermediate reasoning steps involving visual information are often less faithful.
>
> In our work, we classify the intermediate reasoning steps in MCoT based on whether visual information is involved. And we agree that further integrating a more granular classification and analysis on different types (e.g., attribute detection, spatial reasoning, object naming) of intermediate reasoning steps，could provide deeper insights into the faithfulness of MCoT in LVLMs. This could be a promising research direction, potentially categorizing intermediate steps via a third-party LVLM (e.g., GPT-4o like [5]) and analyzing their respective contributions to overall faithfulness.
>
> -----
>
> 5. **W5. Insufficient generalization of CFS to non-autoregressive/encoder-only models.**
> We thank you for your attention to the generalization. This is a valuable consideration for extending the impact of our work. We would like to clarify two key points regarding the scope of our current study:
> - Our work (along with concurrent research) primarily focuses on autoregressive LVLMs, with Qwen2.5-VL-7B as the core experimental model, as it is widely used and has demonstrated strong performance in vision-language tasks.
> - The mentioned models, Flamingo and Kosmos-2.5, do not generate MCoTs, which falls outside the immediate scope of our current work. We appreciate the meaningful suggestion to explore the generalization of CFS to non-autoregressive/encoder-only models in future work. We will also consider adapting and evaluating CFS on such models to assess its effectiveness and generalizability across different LVLM architectures.
>
> [1] Turpin, et al. Language Models Don't Always Say What They Think: Unfaithful Explanations in Chain-of-Thought Prompting. NIPS 2023
>
> [2] Paul, et al. Making Reasoning Matter: Measuring and Improving Faithfulness of Chain-of-Thought Reasoning. EMNLP 2024.
>
> [3] Tanneru, et al. On the Hardness of Faithful Chain-of-Thought Reasoning in Large Language Models. arxiv 2024.
>
> [4] Lanham, et al. Measuring Faithfulness in Chain-of-Thought Reasoning. arxiv 2023.
>
> [5] Xiong, et al. Measuring the Faithfulness of Thinking Drafts in Large Reasoning Models. arxiv 2025.

---

### Official Review · Reviewer_fxZK · 2025-11-02

**Soundness:** 3
**Presentation:** 2
**Contribution:** 2
**Rating:** 4
**Confidence:** 3

**Summary:**

The paper studies the problem of the unfaithfulness of visual-text MCoT in LVLMs. It provides a core finding: irrelevant visual reasoning chains exist in current models and may produce accurate answers. This counterfactual phenomenon indicates that the CoT is dominated by textual cues rather than visual ones. To force MCoT to be visually related, the authors propose a method called the Sufficient-Component Cause Model, which encourages visual evidence to be sufficiently correct and less redundant.The experiments show that the proposed model imporves the faithfulness of LVLMs.

**Strengths:**

- Reasoning faithfulness is an important research topic, especially in the era of black-box LLMs, as it helps improve the interpretability of model behavior and potentially encourages accurate answers.
- The causal intervention analysis provides some insights.
- The paper provides a faithfulness evaluation that is different from previous accuracy-oriented evaluations.

**Weaknesses:**

- Lack of deep analysis of the relationship between accuracy and faithfulness.
    1. First, I want the authors to detail the source of hallucinations in RL learning. The introduction mentions that “inaccurate and insufficient visual information in MCoT may still yield definite, even accurate, answers, suggesting that the MCoT can be unfaithful.” Why? This means the black-box model can perform the task well but does not work in a human-like, step-by-step manner. Why does the accuracy-based RL reward create a bias towards such hallucinations instead of visually related cues? Intuitively, the presence of wrong but interleaved visual cues cannot lead to better rewards. I believe the authors should provide an in-depth explanation of this.
    2. From a quantitative standpoint, the paper mainly provides a faithfulness evaluation but lacks a sufficient analysis of the accuracy (of the final answer). Will faithfulness-based RL training help accuracy? What is the relationship between these two objectives?
- Reliability assessment. As the authors state: “reliability means whether the input visual components are reliable for the model’s prediction.” and “reliability directly reflects the causal consistency between V and A.” However, it is not clear why the prompts in Sec. A.1.2 reflect causal consistency. I suggest the authors provide more explanation on this. Moreover, I believe the causal consistency evaluation is a complex task requiring strong reasoning ability. GPT-4o may not work as well as expert reasoning models like o3/o4. The quality of the assessment should be discussed and evaluated.
- Format inconsistency. For example Sometimes using "Sec. 6.2," sometimes using "Sect."

**Questions:**

see weaknesses

---

> ### Author Response · Authors · 2025-11-24
>
> Thank you for your constructive comments. In response to your concerns, our reply is as follows.
>
> -----
>
> 1. **W1. Lack of deep analysis of the relationship between accuracy and faithfulness.**
>
> - **Q1. Why can "inaccurate and insufficient visual information in MCoT still yield definite, even accurate, answers" (i.e., MCoT unfaithfulness)?**
>
> We appreciate this critical question, as it helps clarify the core problem our work reveals: the unfaithfulness of vision-text MCoT. This hallucination arises because the model relies more heavily on its already robust textual reasoning capabilities during underlying reasoning, while ignoring the visual information in MCoT to produce correct answers. To verify this, we conducted causal analysis via intervention experiments (shown in Table 1). As emphasized in Section 6.2: "*the visual information involved in MCoT may have less impact on the model's underlying reasoning process*,indicating that MCoT exhibits less faithfulness".
>
> - **Q2. Why does the accuracy-based RL reward create a bias towards such hallucinations instead of visually related cues?**
>
> We thank you again for this perceptive question. We kindly refer to the **Sect. 5 (Pitfalls of Existing Methods)** for detailed analysis. Existing RFT methods for MCoT (e.g. Pixel-Reasoner, DeepEyes) only reward two objectives: (1) the final answer accuracy and (2) the format of interleaved vision-text cues. They do **impose no constraints on the correctness or sufficiency of visual components**. This design flaw makes the reward easily hacked through introducing arbitrarily ineffective visual cues and deriving the final (even correct) answer based solely on the textual reasoning, which satisfy the two reward objectievs, and then yield a better reward. Since the reward only cares about "whether visual cues exist" and "whether the answer is correct," the model has no incentive to generate faithful visual information.
>
> - **Q3. Will faithfulness-based RL training help accuracy? What is the relationship between these two objectives?**
>
> We really appreciate your insightful question. Table A5 of Appendix A.3.3 presents a detailed accuracy comparison between our SCCM model and baseline methods. As noted by Reviewer AQe8, "The proposed SCCM method shows substantial improvements on faithfulness metrics but only marginal gains-or even performance drop on overall benchmarks." We acknowledge that the contribution of faithfulness-based RL training to accuracy is not particularly pronounced, and that the relationship between these two objectives remains underexplored. This critical research gap has been identified as a key direction for our future work, where we will delve deeper into balancing these dual goals to achieve more synergistic improvements.
>
> -----
>
> 2. **W2. Reliability assessment.**
>
> - **Q1. It is not clear why the prompts in Sect. A.1.2 reflect causal consistency. More explanation is suggested**
>
> We thank your for your valuable suggestion. The prompts in Sect. A.1.2 is design for reliabiltiy assessments. It explicitly instructing the judger (e.g., GPT-4o) to verify whether V (visual components) could support A (answer), or in other words, whether V is a necessary and non-spurious cause of A. As "reliability directly reflects the causal consistency between V and A", the "Yes" judgment indicates V is a reliable evidence and non-spurious, necessary basis for A, confirming causal consistency; and the "No" judgment signifies the absence of such consistency.
> Additionally, we note that our reliability assessment targets the **stated reasoning process** (i.e., the generated MCoT),  which is complementary to the intervention experiments that focus on the model's **underlying reasoning process**.
>
> - **Q2. GPT-4o may not work as well as expert reasoning models like o3/o4. The quality of the assessment should be discussed and evaluated.**
>
> We appreciate your insightful suggestion. Unfortunately, the accessibility of reasoning models like o3/o4 cannot be obtained in the short term, due to practical constraints. We will take your suggestions into consideration and make improvements and refinements in the future.
>
> -----
>
> 3. **W3. Format inconsistency.**
>
> Thank you for your careful observation and valuable comment on format consistency. We will conduct a full-text review to standardize the format for consistency. Additionally, we will double-check other potential format issues (e.g. citation style suggested by Reviewer AQe8) to ensure the entire manuscript adheres to uniform academic standards.

---

### Author Response · Authors · 2025-12-03
**General Response**

We sincerely thank the reviewers for their valuable feedback!

In this post:
- We summarize and highlight the key strengths of our work as noted by the reviewers.
- We summarize our responses to the main concerns raised by the reviewers.

**Key Strengths Highlighted by Reviewers:**

- **Motivation & Research Value**
  - **Reviewer fxZK**: *"Reasoning faithfulness is an important research topic", "faithfulness evaluation different from previous accuracy-oriented evaluations"*
  - **Reviewer PB4N**: *"addresses the critical question of whether visual explanations are trustworthy,"*
  - **Reviewer AQe8**: *"explores multimodal reasoning faithfulness from a novel and important perspective... largely underexplored in prior work"*

- **Method & Analysis**
  - **Reviewer fxZK**: *"The causal intervention analysis provides some insights"*
  - **Reviewer PB4N**: *"counterfactual interventions inspired by causal inference to isolate intermediate reasoning effects"*
  - **Reviewer qvb5**: *"Reliability and Sufficiency evaluations are interesting, important, valuable"*
  - **Reviewer AQe8**: *"The evaluation pipeline straightforward and methodologically sound", "The intervention-based analyses are particularly insightful"*

- **Presentation & Empirical Support**
  - **Reviewer qvb5**: *"well-written and easy to follow", "core motivation clear with sufficient empirical evidence (e.g., Table 1)"*

**Summary of Responses to Main Concerns:**

1. [To Reviewer **fxZK** and **AQe8**] Faithfulness vs. Accuracy Relationship
    - We acknowledge that faithfulness-based RL training contributes marginally to accuracy, and that the relationship between faithfulness and accuracy requires deeper analysis.
    - We propose a potential explanation for the discrepancy between faithfulness metrics and accuracy improvements: the base model lacks sufficient capability to effectively extract and utilize visual information from the vision-text MCoT.

2. [To Reviewer **fxZK**] The quality of the assessment using GPT-4o
    - We acknowledge the limitation of using GPT-4o for reliability assessment, given the inaccessibility of expert reasoning models like o3/o4.

3. [To Reviewer **fxZK**] Explanation on the prompts of reliability assessment in Sect. A.1.2.
    - We further clarify the prompt and note that our reliability assessment targets the stated reasoning process, complementing the intervention experiments focused on the model's underlying reasoning process.

4. [To Reviewer **PB4N**] Lack of ground truth causal chains
    - We clarify that extracting true causal chains is highly challenging given LVLMs' black-box nature and large parameter scale, and cite related work showing intervention experiments can still yield credible insights without ground-truth causal chains.

5. [To Reviewer **PB4N**] Heuristic and Brittle Counterfactual Generation, unnatural and token-level edits with no semantic change
    - We clarify that the textual counterfactual involves GPT-4o to introduce misleading errors at the sentence level, ensuring naturalness and semantic validity, and visual counterfactual replaces cropped images with random noise, which is in image level and significantly changes the semantics.

6. [To Reviewer **PB4N**] Stochastic behavior of LVLMs
    - We clarify the measures taken to minimize the impact of stochastic behavior of LVLMs: setting the temperature to 0 during inference, and aggregating results from all samples for statistical significance testing.

7. [To Reviewer **PB4N**] Lack of granular analysis on intermediate step types
    - We clarify our work focuses on intermediate steps involving visual information in MCoT, and acknowledge further granular classification and analysis could yield deeper insights.

8. [To Reviewer **qvb5**] Discussion with [1] and evaluation on TreeBench of [1]
    - We detail the comparison between our work and [1], highlighting commonalities and differences in visual cue incorporation, main goals, reward design, and applicability, and have added [1] to the related work section in the revised manuscript.
    - We discuss and compare the mIoU metric used in [1] with our visual sufficiency metric, highlighting their different focuses on spatial accuracy vs. semantic adequacy and will consider adding the evaluation on TreeBench in the next version.

9.  [To Reviewer **AQe8**] Inconsistent results between Table 1 and Table A5.
    -  We clarify that Table 1 only includes vision-text CoT cases; Table A5 covers all cases (including text-only CoT).

10. [To Reviewer **fxZK** and **AQe8**] Format issues: section abbreviation and citation style.
    - We fix them in the revised manuscript.

[1] Wang, et al. Traceable Evidence Enhanced Visual Grounded Reasoning: Evaluation and Methodology. arxiv 2025.

---

### Meta-Review · Area_Chair_Vfgj · 2026-01-03

**Summary:**

This paper studies the faithfulness of multimodal chain-of-thought (MCoT) reasoning in large vision-language models and, via intervention experiments, shows that visual evidence in current MCoT is often ignored or unfaithful despite correct final answers. The paper proposes a new evaluation framework (reliability and sufficiency metrics) and an SCCM learning strategy to encourage sufficient and minimal visual evidence during reasoning.

Reviewers broadly agree that the problem is important and underexplored, and that the intervention-based analyses provide insightful diagnostic evidence of unfaithful visual reasoning. However, reviewers raise substantial concerns about the proposed solution's practical impact and soundness. In particular, multiple reviewers highlight a persistent gap between improved faithfulness metrics and overall task accuracy, including cases where accuracy degrades on vision-text MCoT subsets. Additional concerns include reliance on heuristic counterfactual interventions, lack of ground-truth causal validation, limited evaluation on small-scale models, and unclear generalisation beyond autoregressive MCoT-producing LVLMs.

Overall, while the paper makes a diagnostic contribution, reviewers remain unconvinced that the proposed method sufficiently resolves the identified issues or demonstrates clear downstream benefits.

**Reviewer Concerns:**

**Concerns adequately addressed in the rebuttal:**

* Clarification of why accuracy-based RL rewards can encourage unfaithful visual reasoning by rewarding format rather than correctness or sufficiency of visual cues.
* Explanation of the design and intent of the reliability and sufficiency metrics, and how they complement intervention experiments.
* Clarification of differences between Table 1 and Table A5, including the role of text-only vs. vision-text CoT cases.
* Discussion of stochasticity mitigation (temperature control and statistical testing).
* Comparison with closely related work (e.g., IoU-based visual grounding approaches) and clarification of methodological differences.

**Concerns that remain:**

* The central issue of why improved faithfulness does not translate into improved task accuracy, and in some cases leads to degraded performance, remains unresolved.
* The explanation that the base model lacks the capability to exploit visual information is plausible but largely speculative, lacking quantitative or qualitative evidence.
* The lack of ground-truth causal chains continues to limit confidence in the counterfactual faithfulness evaluation.
* Evaluation is restricted to relatively small models, leaving open whether the observed phenomena persist at larger scales.
* Limited analysis of which types of intermediate reasoning steps are most prone to unfaithfulness, and unclear generalisation to non-MCoT or non-autoregressive models.

**Reviewer Scores:**

* **Reviewer fxZK**: Likely unchanged (marginally below threshold). The rebuttal clarifies motivations and reward design but does not resolve concerns about the faithfulness-accuracy relationship.
* **Reviewer PB4N**: Unchanged (reject). Core concerns about soundness, heuristic interventions, and lack of causal validation remain.
* **Reviewer qvb5**: Likely unchanged or slightly softened but still marginal. Positive about motivation and evaluation, but key suggestions (e.g., additional benchmarks) were deferred.
* **Reviewer AQe8**: Unchanged (marginally below threshold). Despite detailed discussion, the reviewer explicitly maintained their original score due to lack of empirical evidence linking faithfulness improvements to task performance.

---

### Decision · Program_Chairs · 2026-01-26

Reject